# Interferon-β-induced miR-1 alleviates toxic protein accumulation by controlling autophagy

Camilla Nehammer[1,2†], Patrick Ejlerskov[2,3†*], Sandeep Gopal[1], Ava Handley[1], Leelee Ng[1], Pedro Moreira[1], Huikyong Lee[3], Shohreh Issazadeh-Navikas[2], David C Rubinsztein[3,4*], Roger Pocock[1,2*]

[1]Development and Stem Cells Program, Monash Biomedicine Discovery Institute and Department of Anatomy and Developmental Biology, Monash University, Melbourne, Australia; [2]Biotech Research and Innovation Centre, Faculty of Health and Medical Sciences, University of Copenhagen, Copenhagen, Denmark; [3]Cambridge Institute for Medical Research (CIMR), University of Cambridge, Cambridge, United Kingdom; [4]UK Dementia Research Institute, University of Cambridge, Cambridge, United Kingdom

**Abstract** Appropriate regulation of autophagy is crucial for clearing toxic proteins from cells. Defective autophagy results in accumulation of toxic protein aggregates that detrimentally affect cellular function and organismal survival. Here, we report that the microRNA miR-1 regulates the autophagy pathway through conserved targeting of the orthologous Tre-2/Bub2/CDC16 (TBC) Rab GTPase-activating proteins TBC-7 and TBC1D15 in *Caenorhabditis elegans* and mammalian cells, respectively. Loss of miR-1 causes TBC-7/TBC1D15 overexpression, leading to a block on autophagy. Further, we found that the cytokine interferon-β (IFN-β) can induce miR-1 expression in mammalian cells, reducing TBC1D15 levels, and safeguarding against proteotoxic challenges. Therefore, this work provides a potential therapeutic strategy for protein aggregation disorders.

*For correspondence:
patrick.ejlerskov@bric.ku.dk (PE);
dcr1000@cam.ac.uk (DCR);
roger.pocock@monash.edu (RP)

†These authors contributed
equally to this work

Competing interests: The
authors declare that no
competing interests exist.

Reviewing editor: Hitoshi
Nakatogawa, Tokyo Institute of
Technology, Japan

## Introduction

The accumulation of toxic aggregation-prone proteins is a hallmark of multiple human disease states such as Huntington's (HD), Parkinson's (PD), Alzheimer's (AD) and forms of motor neuron disease (*Bosco et al., 2011*). Clearance of aggregation-prone proteins can be promoted by inducing the autophagy pathway (*Ravikumar, 2002*; *Vilchez et al., 2014*). Autophagy is a degradation system that involves sequestration of cytoplasmic proteins and organelles by double-layered membranes that form vesicles called autophagosomes. Fusion of autophagosomes with lysosomes results in degradation of their contents and thereby removes toxic proteins and damaged organelles from cells to maintain homeostasis. Due to the central role of autophagy in the removal of aggregation-prone proteins, a better understanding of mechanisms controlling autophagy is essential for the identification of novel therapeutic opportunities for multiple disease states.

microRNA (miRNAs) are single-stranded, non-coding RNAs of ~21–24 nucleotides in length that post-transcriptionally regulate the expression of target genes (*Bartel, 2009*; *Krol et al., 2010*). miR-NAs predominantly interact with mRNA targets through imperfect binding to motifs in target mRNA 3'-untranslated regions (3'UTRs) (*Bartel, 2009*). miRNA:mRNA interactions negatively impact the stability and translational capacity of mRNA targets in a rapid and reversible manner (*Fabian et al., 2010*). The nature of imperfect binding specificity means that a single miRNA can regulate a large number of mRNA targets involved in complex cellular processes, thereby tightly controlling genetic networks during development and in response to stress (*Pocock, 2011*). As such, dysregulation of

miRNA-controlled processes can cause severe physiological consequences for animal behavior and survival (*Boulias and Horvitz, 2012*; *de Lencastre et al., 2010*; *Finger et al., 2019*; *Grueter et al., 2012*; *Kagias and Pocock, 2015*; *Nehammer et al., 2015*; *Vora et al., 2013*). In addition, due to their rapid and reversible regulatory capacity, miRNAs are prime candidate facilitators of responses to proteotoxic stress.

miR-1 is a highly conserved miRNA that is detected in muscle, neurons and circulatory body fluid of multiple metazoan phyla (*de Rie et al., 2017*; *Kopkova et al., 2018*; *Kusuda et al., 2011*; *Sokol, 2012*) (*Figure 1A*). Multiple roles of miR-1 have been identified in the development and function of muscle in multiple systems (*Simon et al., 2008*; *Sokol and Ambros, 2005*; *Zhao et al., 2005*). In *C. elegans*, *mir-1* is expressed in pharyngeal and body wall muscle (BWM) and regulates retrograde signaling at neuromuscular junctions of the latter (*Hu et al., 2012*; *Simon et al., 2008*). However, the function of miR-1 in stress responses, including autophagy, is poorly understood. Intriguingly, miR-1 expression is depleted in a *Drosophila melanogaster* model of AD (*Kong et al., 2014*) and human miR-1 is reduced in the cerebrospinal fluid of patients with PD (*Gui et al., 2015*; *Margis et al., 2011*). This prompted us to investigate if *mir-1* is required for preventing the accumulation of aggregation-prone proteins.

Here, we reveal an important function of miR-1 in combatting multiple proteotoxic threats. We identify a highly conserved pathway through which miR-1 controls the accumulation of toxic protein aggregates through the autophagy pathway in *C. elegans* and mammalian cells. The key regulatory mechanism by which miR-1 controls toxic protein accumulation is through direct control of the Tre-2/Bub2/CDC16 (TBC) Rab GTPase-activating proteins (Rab GAPs) TBC-7 and TBC1D15 in *Caenorhabditis elegans* and mammalian cells, respectively. In concurrence with previous in vitro and in vivo studies, we found that TBC1D15 specifically functions as a Rab GAP for the small GTPase Rab7 - a known regulator of autophagy (*Gutierrez et al., 2004*; *Peralta et al., 2010*; *Zhang et al., 2005*). As such, we show that TBC1D15 reduces the amount of active GTP-bound Rab7 and a constitutive active Rab7 mutant circumvents the TBC1D15-mediated block in autophagy. In agreement with this mechanistic association, we found that proper regulation of TBC protein expression by miR-1 permits appropriate autophagic flux and clearance of toxic protein aggregates. Finally, we discover that the cytokine interferon-β (IFN-β) positively regulates miR-1 expression in mammalian cells to promote autophagy and clearance of toxic protein aggregates. Together, these findings suggest novel therapeutic approaches to prevent and/or clear toxic protein aggregation through the autophagy pathway.

## Results

### *mir-1* prevents polyglutamine aggregation

To gain insight into whether *mir-1* potentially controls protein aggregation, we assayed *mir-1* function using an established *C. elegans* transgenic polyglutamine model, which has the same type of mutation as seen in HD (*Figure 1*) (*Morley et al., 2002*). This model expresses a polypeptide of 40 glutamine residues fused to yellow fluorescent protein (YFP) in BWM (Q40::YFP), hereafter referred to as Q40 (*Morley et al., 2002*). Using two independently-derived *mir-1* deletion alleles, *mir-1 (gk276)* and *mir-1(n4102)*, we found that loss of *mir-1* increases Q40 accumulation in BWM, without affecting Q40 expression levels (*Figure 1B–D* and *Figure 1—figure supplement 1*). This phenotype was not due to a non-specific change in muscle cell miRNA profile, as loss of the muscle-specific *mir-80*, did not affect Q40 aggregation (*Figure 1D*). In *C. elegans*, *mir-1* is expressed in BWM and the pharynx (*Simon et al., 2008*). To characterize the functional locale of *mir-1* in regulating Q40 aggregation, we performed tissue-specific rescue experiments. We found that *mir-1* expression in BWM, but not in the pharynx or intestine, rescued the aberrant Q40 aggregation phenotype in *mir-1 (gk276)* animals (*Figure 1E*), demonstrating that *mir-1* acts cell autonomously to control Q40 accumulation. miRNAs predominantly regulate gene expression through imperfect base-pairing with target mRNA 3′UTRs, causing RNA instability and/or translational repression (*Bartel, 2009*; *Lewis et al., 2005*). To determine if a canonical miRNA:mRNA target interaction is required for *mir-1* function, we mutated two conserved nucleotides in the *mir-1* seed sequence and repeated the rescue experiment (*Figure 1E*). Expressing mutated *mir-1* (*mir-1\**) in BWM failed to rescue the Q40 protein aggregation phenotype of *mir-1(gk276)* animals (*Figure 1E*).

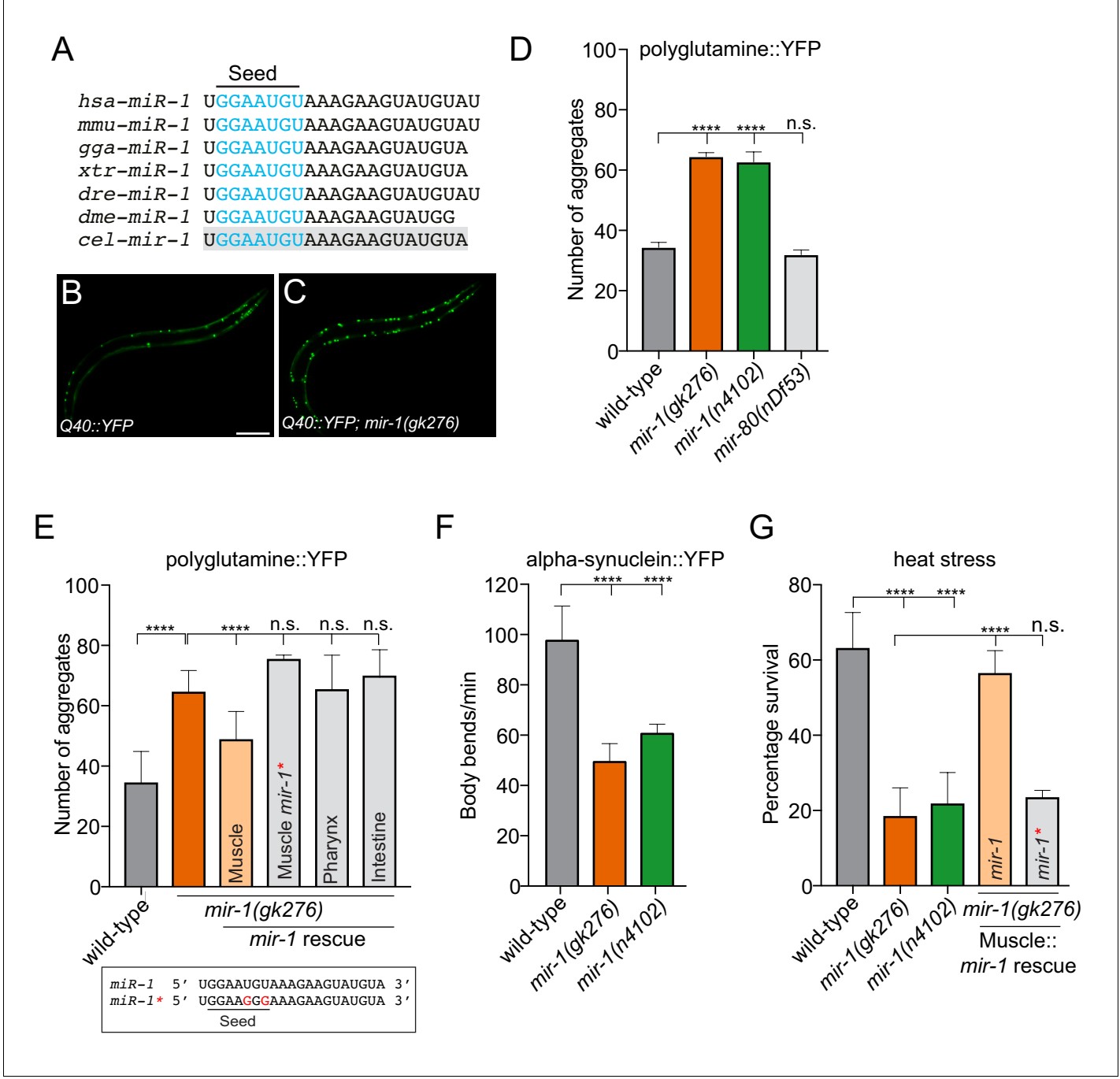

**Figure 1.** *mir-1* protects against proteotoxic stress. (A) Alignment of mature *miR-1* sequences indicates deep conservation. The seed sequence of each *miR-1* family member is highlighted in blue and the conservation of *C. elegans mir-1* is highlighted in gray. *hsa = Homo sapiens, mmu = Mus musculus, gga = Gallus gallus, xtr = Xenopus tropicalis, dre = Danio rerio, dme = Drosophila melanogaster, cel = Caenorhabditis elegans.* (B–C) Visualization of Q40::YFP aggregates (green foci) in (B) wild-type and (C) *mir-1(gk276)* animals. Scale bar, 50 μm. (D) Quantification of Q40::YFP aggregation in wild-type, *mir-1(gk276), mir-1(n4102)* and *mir-80(nDf53)* animals. (E) Quantification of Q40::YFP aggregates in wild-type, *mir-1(gk276)* and *mir-1(gk276)* animals transgenically-expressing the *mir-1* hairpin in body wall muscle (*myo-3* promoter), pharynx (*myo-2* promoter) or intestine (*ges-1* promoter). Mutation of the *mir-1* seed sequence (Muscle *mir-1\**) abrogates rescue from body wall muscle. Mature *mir-1* sequences (wild-type *mir-1* or mutated *mir-1\**) used for rescue experiments are shown (box). Red nucleotides indicate the mutations in the seed sequence used in *mir-1\** rescue experiments, which are predicted to hinder interactions with *mir-1* targets. (F) Body bends in wild-type, *mir-1(gk276)* and *mir-1(n4102)* mutant animals expressing α-synuclein::YFP. (G) Survival of wild-type, *mir-1(gk276)* and *mir-1(n4102)* animals after exposure to 4 hr of 35°C heat stress. Transgenic expression of wild-type *mir-1* hairpin, but not mutated *mir-1\**, in body wall muscle rescues *mir-1(gk276)* heat stress sensitivity. All experiments were performed in triplicate

*Figure 1 continued on next page*

Figure 1 continued

and at least 10 animals were scored per experiment. Error bars show standard error of the mean (SEM). ****p<0.0001, n.s. not significant to the control (one-way ANOVA analysis, followed by Dunnett's multiple comparison test).

The online version of this article includes the following figure supplement(s) for figure 1:

**Figure supplement 1.** Quantification of Q40::YFP Expression.
**Figure supplement 2.** Motility Analysis.
**Figure supplement 3.** *mir-1* prevents the formation of α-synuclein inclusions.
**Figure supplement 4.** *mir-1* Lifespan Analysis.

The expression of expanded polyglutamine repeats in muscle is toxic and progressively affects muscle function and *C. elegans* motility (*Morley et al., 2002*). We found that Q40 toxicity was exacerbated in *mir-1* mutant animals in an age-dependent manner (*Figure 1—figure supplement 2*). To determine if *mir-1* has an effect on motility in a non-proteotoxic environment, we tested the motility of animals expressing the control Q0::YFP transgene as well as wild-type and *mir-1(gk276)* animals devoid of transgenes (*Figure 1—figure supplement 2*). We observed a slight decrease in motility in *mir-1* mutant animals expressing Q0::YFP and with no transgene (*Figure 1—figure supplement 2*), suggesting that loss of *mir-1* may cause defects in muscle proteostasis, which are exacerbated when animals are overloaded with protein aggregates. Taken together, *mir-1* prevents Q40 protein accumulation in BWM and protects against proteotoxicity, presumably through 3'UTR-directed regulation of its target gene(s).

## *mir-1* protects against proteotoxic threats

To determine whether *mir-1* generally guards against proteotoxic stress, we examined two other stress paradigms. First, we used a PD model where the aggregation-prone human α-synuclein is fused to YFP and transgenically expressed in BWM (*van Ham et al., 2008*). This model elicits age-dependent accumulation of α-synuclein inclusions and a decline in motility (*Cooper et al., 2015*; *van Ham et al., 2008*). We found that loss of *mir-1* causes an increase in the number of α-synuclein:: YFP inclusions and ~50% reduction in motility (*Figure 1F* and *Figure 1—figure supplement 3*). Second, we analysed *mir-1* function in heat stress sensitivity - a more general proteotoxic stress (*Figure 1G*). Elevated temperature places added pressure on the protein folding machinery causing endogenous proteins to misfold and form toxic aggregates (*Wallace et al., 2015*). We found that loss of *mir-1* caused severe heat stress sensitivity and that resupplying wild-type *mir-1*, but not mutated *mir-1* (*mir-1\**), in BWM rescues this phenotype (*Figure 1G*). In addition to acute environmental stressors, the aging process causes accumulation of misfolded proteins (*Brignull et al., 2007*). Surprisingly, *mir-1* mutant animals exhibit wild-type lifespan (*Figure 1—figure supplement 4*), suggesting that *mir-1* primarily acts to combat proteotoxic challenges and/or that parallel pathways overcome proteostasis defects during aging. Alternatively, the activities of *mir-1* in controlling protein aggregation are uncoupled from lifespan regulation. Together, our data show that *mir-1* plays a broad role in protecting against the accumulation of aggregation-prone proteins and the toxic effect of acute heat stress.

## *mir-1* Directly Regulates *tbc-7* Expression in *C. elegans*

Our data provide evidence that *mir-1* targets an mRNA or mRNAs that encode vital regulators of proteotoxic stress. To identify these targets, we employed two complementary approaches. We used RNA sequencing to identify differentially expressed genes in *mir-1(gk276)* animals compared to wild-type (*Supplementary files 1–2*). In parallel, we knocked down the expression of predicted *mir-1* target genes (TargetScanWorm release 6.2) using RNA-mediated interference (RNAi) to identify regulators of *mir-1(gk276)* heat stress sensitivity (*Figure 2—figure supplement 1*). These experiments revealed a single gene called *tbc-7* as a putative candidate *mir-1* target. We found that *tbc-7* mRNA, a highly conserved predicted *mir-1* target, is elevated in *mir-1(gk276)* animals (*Figure 2A* and *Supplementary files 1–2*). Further, reducing *tbc-7* expression fully suppressed heat stress sensitivity of *mir-1(gk276)* animals (*Figure 2B* and *Figure 2—figure supplement 1*). TBC-7 is uncharacterized and predicted to encode a Rab GTPase-activating protein (Rab GAP) member of the Tre-2/Bub2/CDC16 (TBC) family (*Gao et al., 2008*). To determine the *tbc-7* expression pattern, we

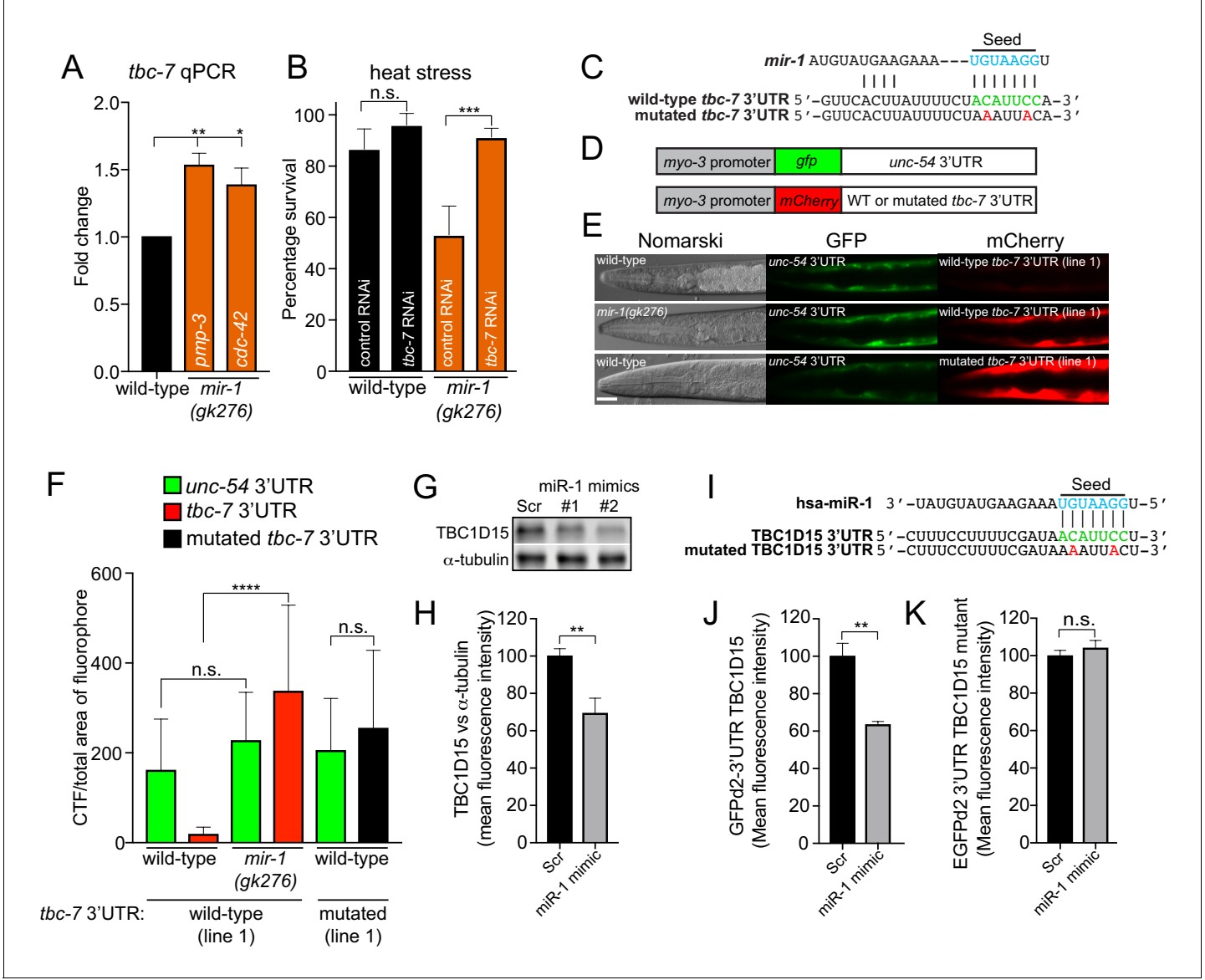

**Figure 2.** *miR-1* directly regulates TBC 3'UTRs in *C. elegans* and mammals. (**A**) Relative *tbc-7* mRNA levels measured by quantitative real-time PCR in L4 larvae. Data normalized to values for wild-type worms. Two independent reference genes (*pmp-3* and *cdc-42*) were used. Error bars show standard error of the mean (SEM) obtained from n = 3 biological replicates and three technical replicates each. **p<0.001, *p<0.005 (one-way ANOVA analysis, followed by Dunnett's multiple comparison test). (**B**) Survival of wild-type and *mir-1(gk276)* animals (incubated on control (L4440) or *tbc-7* RNAi bacteria) after exposure to 4 hr of heat stress (35°C) (n = 30). ***p<0.001, n.s. not significant (one-way ANOVA analysis, followed by Dunnett's multiple comparison test). (**C**) Predicted *mir-1* binding site on the 3'UTR of *tbc-7* mRNA (green) and seed sequence in *mir-1* (blue). Mutated nucleotides in the *tbc-7* 3'UTR for experiments (**E–F**) are in red. (**D**) Indicated DNA constructs were co-transformed as multi-copy extrachromosomal arrays for experiments in (**E–F**). (**E**) Expression of heterologous reporter transgenes for control *unc-54* 3'UTR (*gfp*) and wild-type and mutated *tbc-7* 3'UTR (*mCherry*) constructs in body wall muscle. (**F**) Quantification of *gfp* and *mCherry* fluorescence of transgenic animals calculated as CTF/total area of fluorophore (n = 30). ****p<0.0001, n.s. not significant (one-way ANOVA analysis, followed by Dunnett's multiple comparison test). (**G**) WB of TBC1D15 and α-tubulin and (**H**) quantified bands from HeLa cells transfected with scrambled (Scr) or miR-1 mimics (n = 5). Data are mean fluorescence intensities ± SEM. **p<0.01 (Students t-test). (**I**) Predicted miR-1 binding site on the 3'UTR of TBC1D15 mRNA (green) and seed sequence in miR-1 (blue). Mutated nucleotides in the TBC1D15 3'UTR for experiments (**L–M**) are in red. (**J–K**) Quantification of flow cytometry analysis of HeLa cells co-expressing scrambled (Scr) or miR-1 mimic together with (**J**) GFPd2-3'UTR TBC1D15 (n = 4) or (**K**) mutated GFPd2-3'UTR TBC1D15$_{mutant}$ (n = 5). Data are mean fluorescence intensities ± SEM, **p<0.01 (Students t-test).

The online version of this article includes the following figure supplement(s) for figure 2:

**Figure supplement 1.** RNAi screen to identify *mir-1* targets important for the heat stress response.

**Figure supplement 2.** *tbc-7* expression pattern.

*Figure 2 continued on next page*

*Figure 2 continued*

**Figure supplement 3.** Overexpression of *tbc-7* causes Q40::YFP aggregation.
**Figure supplement 4.** *miR-1* targeting of TBC proteins is conserved.
**Figure supplement 5.** TBC1D15 3′UTR analysis.

generated a transgene driving green fluorescent protein (GFP) under the control of a *tbc-7* promoter (*Ptbc-7::gfp*). We detected fluorescence in the intestine, head cells and importantly in BWM (***Figure 2—figure supplement 2***), the site-of-action for *mir-1* in regulating proteotoxic stress. Together, our data confirm single-cell transcriptional profiling that detected *tbc-7* expression in multiple tissues, including in BWM (***Cao et al., 2017***).

To assess whether *mir-1* directly regulates *tbc-7* expression in vivo, we used a well-established 3′UTR sensor assay (***Pedersen et al., 2013***). We generated a transgene expressing two reporters in BWM: one red fluorescent protein (*mCherry*) 'sensor' reporter controlled by the *tbc-7* 3′UTR and another with a *gfp* 'control' reporter controlled by the *unc-54* 3′UTR, which does not contain any *mir-1* binding sites (***Figure 2C–F***). In wild-type animals expressing this transgene, we detected robust *gfp* expression and weak *mCherry* expression (***Figure 2E–F***). When the same transgene was transferred into the *mir-1(gk276)* mutant, we observed high *mCherry* expression, suggesting that *mir-1* directly represses the *tbc-7* 3′UTR (***Figure 2E–F***). Further, when we disrupted the predicted *mir-1* binding site in the *tbc-7* 3′UTR *mCherry* sensor, we observed high red fluorescence in wild-type animals, confirming that *mir-1* regulation is required to repress *tbc-7* expression (***Figure 2E–F***). As *mir-1* directly downregulates *tbc-7* expression, we hypothesized that overexpressing *tbc-7* in BWM (the site of *mir-1* action) would phenocopy a *mir-1* mutant phenotype in wild-type animals. We found that overexpressing *tbc-7* in wild-type BWM causes increased accumulation of Q40 aggregates, but did not enhance the Q40 aggregation phenotype of *mir-1(gk276)* animals (***Figure 2—figure supplement 3***).

## miR-1 Directly Regulates TBC1D15 Expression in Mammals

The mature *mir-1* sequence is completely conserved from worms to humans (***Figure 1A***). We found that *Drosophila* and human orthologs of TBC-7 - Skywalker and TBC1D15, respectively - are also predicted targets of miR-1 (***Figure 2—figure supplement 4***), and in the case of TBC1D15, this relationship is suggested by CLIP data (***Kishore et al., 2011***). We have shown that *mir-1* directly regulates *tbc-7* expression in *C. elegans.* Therefore, to ask if the regulatory function of miR-1 is conserved, we measured TBC1D15 protein in mammalian cells (***Figure 2G–H***). We found that miR-1 overexpression reduced levels of TBC1D15 mRNA and protein in HeLa cells (***Figure 2G–H*** and ***Figure 2—figure supplement 5A***). Additionally, a *gfp* 'sensor' reporter containing the wild-type TBC1D15 3′UTR is downregulated by miR-1 overexpression, and this downregulation requires the miR-1 binding site (***Figure 2I–K*** and ***Figure 2—figure supplement 5B–C***). These data show that, as in *C. elegans*, miR-1 directly interacts with the 3′UTR of a TBC protein-encoding mRNA to downregulate its expression.

## miR-1 Regulation of TBC Protein Levels Controls Autophagy

Our data reveal a miR-1/TBC protein regulatory axis is conserved in worms and mammals. TBC proteins control vesicular transport in cells by enhancing Rab GTPase hydrolysis of guanosine triphosphate (GTP) to guanosine diphosphate (GDP) (***Strom et al., 1993***). Rab GTPase guanosine-binding status is important for interaction-specificity with effector molecules (***Stein et al., 2012***). Therefore, TBCs can precisely control the specificity and rate of vesicular transport routes and thus have been functionally associated with autophagy (***Kern et al., 2015***). Indeed, TBC1D15 acts as a Rab GAP for the small GTPase Rab7 - a known autophagy regulator (***Gutierrez et al., 2004***; ***Kirisako et al., 1999***; ***Peralta et al., 2010***; ***Zhang et al., 2005***). This posits an evolutionary conserved function for miR-1 in controlling autophagy through TBC protein regulation.

To examine the function of *mir-1* and *tbc-7* in autophagy we used two independent *C. elegans* strains. First, we used a strain that expresses a GFP-tagged LGG-1/Atg8 reporter, to enable visualization of autophagosomes as fluorescent puncta (***Figure 3***) (***Chang et al., 2017***; ***Kang et al., 2007***). When autophagy is activated, cytosolic LGG-1-I/Atg8 is conjugated to phosphatidylethanolamine at

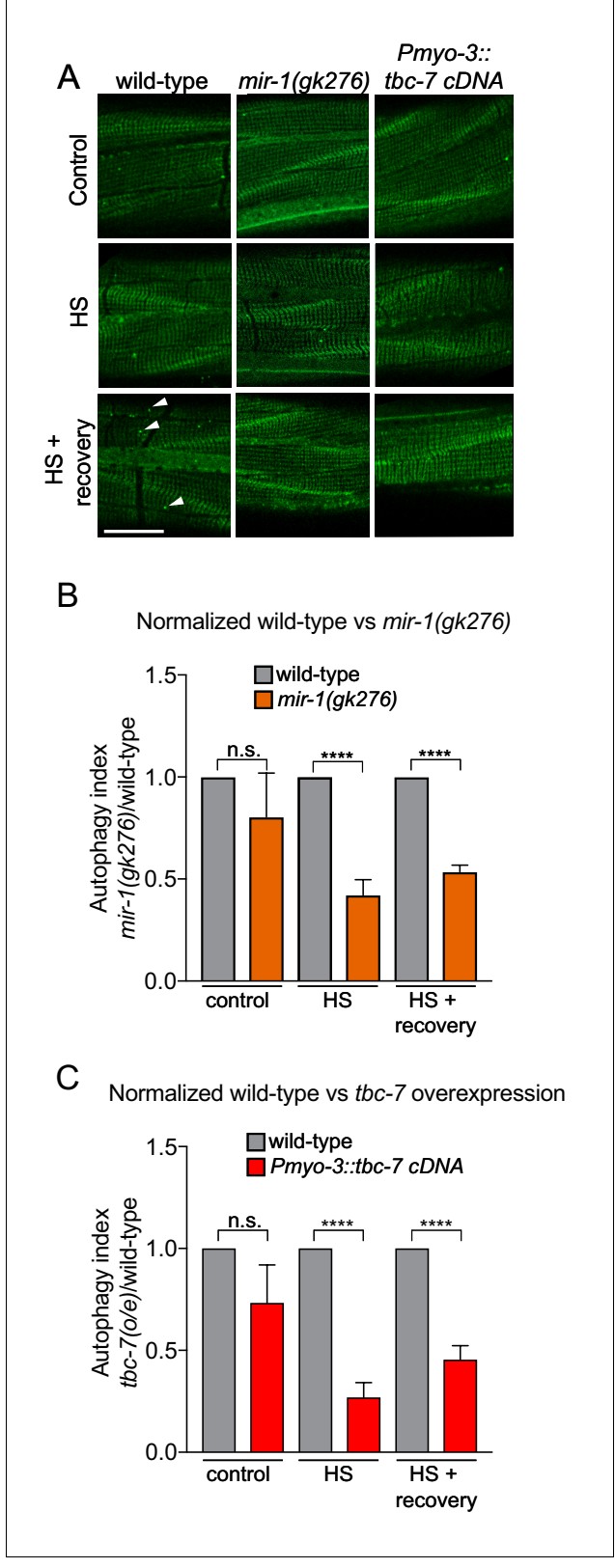

**Figure 3.** *mir-1* and *tbc-7* control stress-induced autophagy. (**A**) Fluorescent images of BWM expressing GFP::LGG-1/Atg8 in wild-type, *mir-1(gk276)* and *Pmyo-3::tbc-7* overexpressing animals under control conditions, immediately after heat shock for 1 hr at 35°C (HS) or 1 hr after recovery from heat shock at 15°C (HS + recovery). GFP::LGG-1 puncta = arrowheads. Scale bar, 10 μm. (**B–C**) Quantification of GFP::LGG-1/Atg8 puncta in

*Figure 3 continued on next page*

*Figure 3 continued*
BWM of animals and conditions shown in (**A**). The values represent the number of green puncta in *mir-1(gk276)* (**B**) and *Pmyo-3::tbc-7* overexpressing (**C**) animals normalized to one green puncta in wild-type animals for each condition. n > 15. ± SEM ****p<0.0001, n.s. not significant (Welch's t-test).
The online version of this article includes the following figure supplement(s) for figure 3:

**Figure supplement 1.** mir-1 controls stress-induced autophagy.

the phagophore membrane, forming lipidated LGG-1-II/Atg8, that is present in vesicular autophagosome structures (*Bento et al., 2016*). Thus, the number of GFP::LGG-1-positive puncta can be used as a readout of autophagic activity. We found that the number of GFP::LGG-1 puncta in *mir-1 (gk276)* mutant BWM was not different from wild-type in standard laboratory conditions (*Figure 3*). However, loss of *mir-1* abrogates the autophagic heat stress response, as does *tbc-7* overexpression in BWM (*Figure 3*). To confirm these data, we used an alternative strain (mCherry::GFP::LGG-1) that enables autophagic flux to be examined by measuring autolysosome number (*Chang et al., 2017*). Using this reporter, we found that *mir-1* mutant animals have a reduced number of autolysosomes in unstressed conditions and are unable to mount an autophagic response to heat stress (*Figure 3— figure supplement 1*). Together, these data suggest that *mir-1* regulation of *tbc-7* expression is required to control autophagy-dependent stress responses.

To examine the role of miR-1/TBC1D15 on autophagy in mammalian cells, we measured LC3-positive vesicular structures and LC3-II protein abundance (*Figure 4*). Overexpression of miR-1 in HeLa cells increased both the number of LC3-positive puncta per cell and total LC3-II protein levels, indicating an increase in the number of autophagosomes (*Figure 4A–B* and *Figure 4—figure supplement 1A*). To investigate autophagy flux further we made use of the vesicular ATPase inhibitor bafilomycin A1 (BafA1), which inhibits lysosomal acidification and thereby blocks the ability of lysosomes to fuse with autophagosomes. This enables assessment of autophagy flux indicated by the amount of LC3-II positive autophagosomes accumulating over a time period of 4 hr (*Rubinsztein et al., 2009*). Upon expression of miR-1 mimics in combination with Baf A1, LC3-II levels were even further increased, suggesting that miR-1 promotes autophagy flux (*Figure 4A–B*). TBC1D15 knockdown cells exhibited the same phenotype as miR-1 overexpression (*Figure 4—figure supplement 1B–C*), suggesting that miR-1-mediated downregulation of TBC1D15 promotes autophagy flux. To support this, miR-1 overexpression also increases the autolysosome/autophagosome ratio, scored with a mRFP-GFP-LC3 reporter (*Figure 4—figure supplement 2A–C*), confirming that this is indeed due to an increase in autophagy flux rather than a blockage of the pathway.

We next overexpressed TBC1D15 in HeLa cells to further characterize its role in autophagy. We found that TBC1D15 overexpression increased LC3-II levels, which did not further increase in the presence of Baf A1, indicating a block in the autophagy pathway (*Figure 4C–D*). This was further validated by TBC1D15 overexpression in HeLa cells expressing the mRFP-GFP-LC3 reporter, which revealed large stationary autophagosomes and decreased autolysosome/autophagosome ratio (*Figure 4E–G* and *Videos 1–2*). miR-1 did not change basal LC3-II levels in cells overexpressing TBC1D15, in the presence or absence of Baf A1, when compared to cells expressing TBC1D15 with scrambled control (*Figure 4H–I*). Thus, ectopic expression of TBC1D15, which is not regulated by its endogenous 3'UTR, masks miR-1-induced autophagy flux presumably via its blocking effect on autophagy. Together, these data show that miR-1 regulation of TBC1D15 controls autophagy and that unrestricted expression of TBC1D15 causes a late-stage block in autophagy flux.

## miR-1 Prevents Mutant Huntingtin Aggregation by Controlling Autophagy

Our collective data suggest that manipulation of the miR-1/TBC1D15 axis could be used to reduce the accumulation of polyglutamine aggregates in human cells. To examine this possibility, we expressed EGFP-tagged mutant human huntingtin exon 1 fragment with 74 polyQ repeats (HTT$_{Q74}$) in HeLa cells and manipulated miR-1 and TBC1D15 levels (*Figure 5*). Overexpression of two independently derived miR-1 mimics reduced the number of cells containing HTT$_{Q74}$ aggregates after 48 hr of HTT$_{Q74}$ expression (*Figure 5A* and *Figure 5—figure supplement 1*), a phenomenon that correlates with autophagy induction (*Ravikumar, 2002*). Conversely, miR-1 has no effect on the

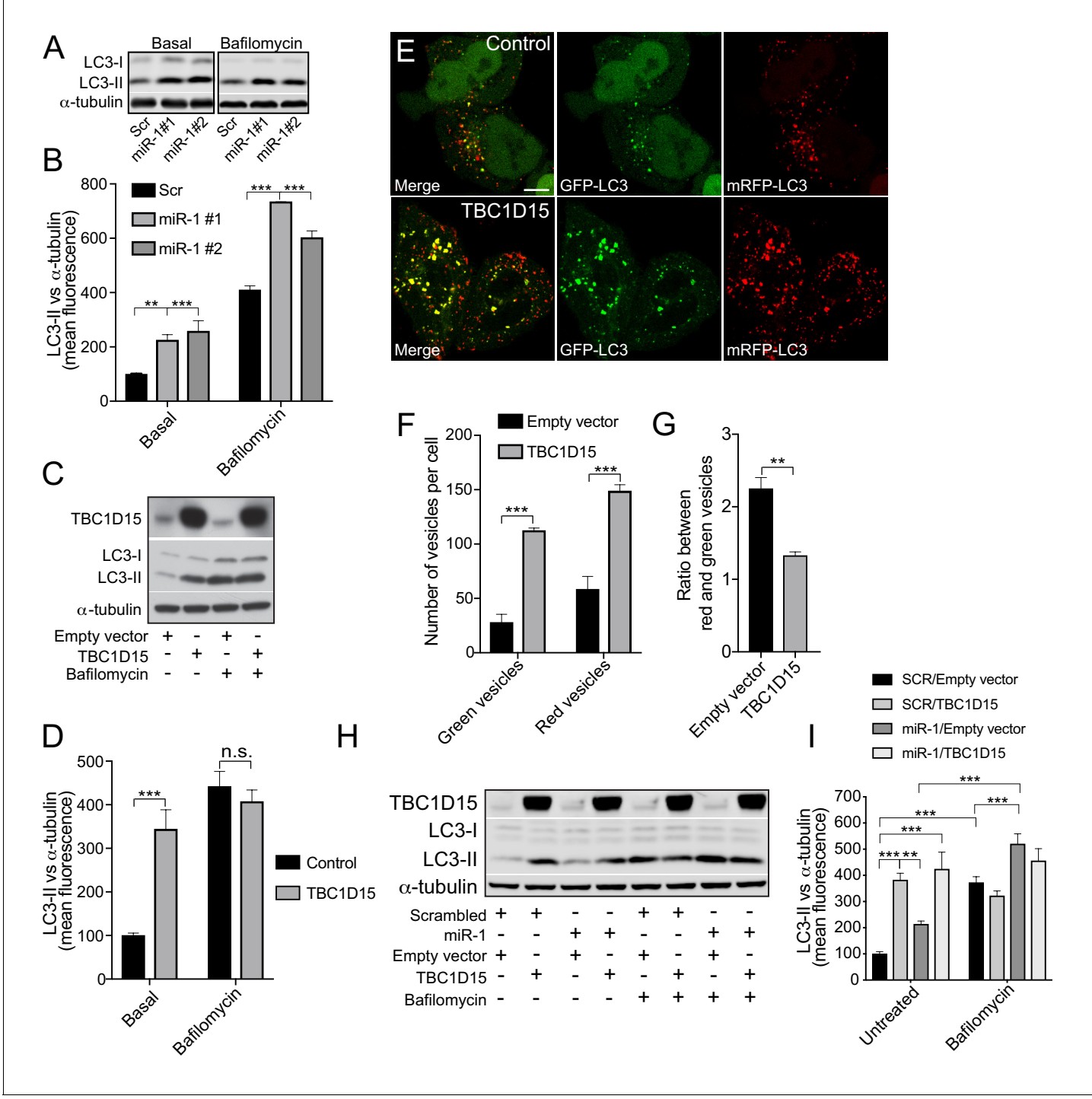

**Figure 4.** Human miR-1 regulates autophagy by controlling TBC1D15 expression. (**A**) WB and (**B**) quantification of LC3-II normalised to α-tubulin from HeLa cells expressing Scr or miR-1 mimics +/- bafilomycin. Data are mean fluorescence intensities of bands ± SEM (n = 3–5). **p<0.01, ***p<0.001 (one-way ANOVA with Dunnett's correction). (**C**) WB and (**D**) quantification of LC3-II normalised to α-tubulin from HeLa cells expressing empty vector (control) or TBC1D15 overexpression vector +/- bafilomycin. Data are mean fluorescence intensities of bands ± SEM normalised to α-tubulin (n = 5). n.s. not significant to the control, ***p<0.001 (two-way ANOVA with Bonferroni correction). (**E**) IF images of HeLa cells stably expressing mRFP-GFP-LC3 and transfected with empty vector (control) or TBC1D15 overexpression vector. Scale bar, 10 μm. (**F**) Quantification of green and red vesicles and (**G**) red/green vesicle ratio from (**E**) ± SEM (n = 3, 12–14 cells per replicate). **p<0.01, ***p<0.001 (Student's t-test). (**H**) WB and (**I**) quantification of HeLa cells co-transfected with Scr or miR-1 mimic together with empty vector or TBC1D15 overexpression vector +/- bafilomycin. Data are mean fluorescence intensities of LC3-II bands normalized to α-tubulin ± SEM (n = 7). **p<0.01, ***p<0.001 (two-way ANOVA with Bonferroni correction).

*Figure 4 continued on next page*

**Figure supplement 1.** miR-1 and TBC1D15 control autophagy.

percentage of cells with $HTT_{Q74}$ aggregates in autophagy-null cells, using CRISPR/Cas9 knockout of ATG16L1 - a protein essential for autophagosome formation - confirming that miR-1 acts through the autophagy pathway (**Figure 5B**). We further found that TBC1D15 knockdown lowered, and overexpression increased, the percentage of $HTT_{Q74}$ positive cells (**Figure 5C–D**). The combination of TBC1D15 overexpression and $HTT_{Q74}$ induced considerable cell death in the HeLa cells after 48 hr of expression, thus for this condition the transgenes were only expressed for 24 hr. Co-expression of miR-1 had no effect on the percentage of cells with $HTT_{Q74}$ aggregates induced by TBC1D15 overexpression (**Figure 5E**) for similar reasons as in **Figure 4H–I** (see above), showing that correct regulation of TBC1D15 is important to prevent the accumulation of toxic protein aggregates.

## IFN-β induces miR-1 expression to prevent mutant huntingtin aggregation

We next examined the therapeutic potential of boosting miR-1 expression to reduce $HTT_{Q74}$ accumulation through the autophagy pathway. By examining the extant literature, we discovered that the cytokine interferon-β (IFN-β) positively regulates miR-1 expression in hepatic cells (**Pedersen et al., 2007**). As IFN-β can promote autophagy flux (**Ambjørn et al., 2013**) and alleviate models of neurodegenerative disease (**Ejlerskov et al., 2015**), we hypothesized that the therapeutic effect of IFN-β may, at least in part, be due to induction of miR-1. In mouse primary cortical neurons,

we found that IFN-β induced miR-1 expression by 2-fold and concomitantly decreased Tbc1d15 protein levels (**Figure 6A–C**). Conversely, brains of 3 month old *Ifnb*$^{-/-}$ mice have significantly increased levels of Tbc1d15 protein (**Figure 6— figure supplement 1A–B**). To examine whether IFN-β can induce autophagy and reduce $HTT_{Q74}$ accumulation through miR-1/TBC1D15, we used HeLa cells as a model. We found that IFN-β also induces miR-1 and reduces TBC1D15 levels in HeLa cells (**Figure 6—figure supplement 2**). We found that IFN-β regulation of TBC1D15 requires an intact miR-1 3′UTR binding site as a wild-type TBC1D15 3′UTR *gfp* sensor, but not a miR-1 binding site-mutated TBC1D15 3′UTR *gfp* sensor, is downregulated by IFN-β (**Figure 6D–E**). To establish if IFN-β depends on miR-1 to control autophagy and $HTT_{Q74}$ accumulation, we performed genetic knockdown and overexpression experiments. First, we found that IFN-β treatment reduces $HTT_{Q74}$ aggregate accumulation, but this was abolished in cells stably expressing a hairpin inhibitor against miR-1 (Off-miR-1) (**Figure 6F–G**). In the presence of Baf A1, IFN-β causes a further increase in LC3-II levels, however, this is diminished in Off-miR-1 HeLa cells (**Figure 6H–I**). These data indicate that IFN-β enhancement of autophagy flux and reduction of $HTT_{Q74}$ accumulation is dependent on miR-1 induction.

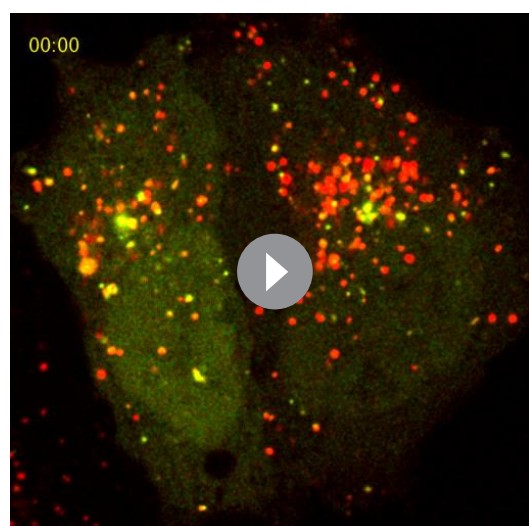

**Video 1.** Autophagy flux in cells expressing an empty (control) vector. HeLa cells stably expressing mRFP-GFP-LC3 were transfected with empty vector (**Video 1**) or TBC1D15 overexpression vector (**Video 2**) and live cell imaging was conducted the following day. Notice the presence of large immobile mRFP- and GFP-positive autophagosomes in TBC1D15 overexpressing cells, implying a block in autophagosome maturation. Cells were imaged once every second for a period of 2 min and the movies are displayed at a speed of 10 frames per second.

https://elifesciences.org/articles/49930#video1

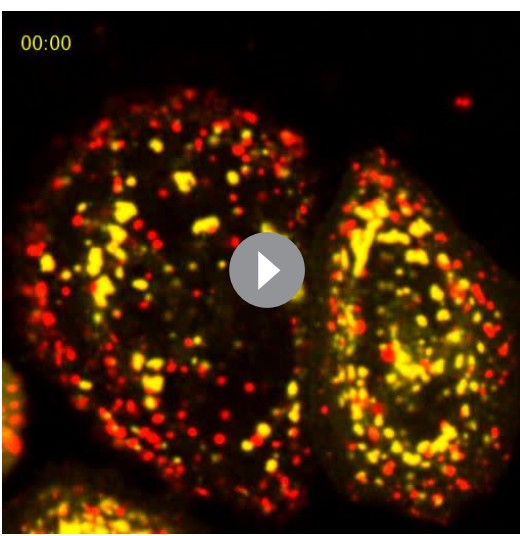

**Video 2.** TBC1D15 overexpression causes large stationary autophagosomes.
https://elifesciences.org/articles/49930#video2

We have shown that the beneficial effects of miR-1 overexpression on autophagy and $HTT_{Q74}$ accumulation is abrogated by an autophagy block caused by TBC1D15 overexpression. We therefore tested whether disruption of autophagy flux by TBC1D15 would prevent IFN-β promotion of autophagy. Treating control cells with either IFN-β or Baf A1 caused an increase in LC3-II levels (*Figure 6—figure supplement 3A–B*). Additionally, co-treatment with IFN-β and Baf A1 generates a further increase in LC3-II levels compared to either IFN-β or Baf A1 alone, supporting the role of IFN-β in promoting autophagy flux (*Figure 6—figure supplement 3A–B*). As shown previously, TBC1D15 overexpression blocks autophagy (*Figure 4D*). Neither IFN-β nor Baf A1, either independently or in combination, further increased LC3-II levels caused by TBC1D15 overexpression, further confirming that excess TBC1D15 causes a late-stage autophagy block (*Figure 6—figure supplement 3A–B*). Correct regulation of TBC1D15 is important, as TBC1D15 overexpression abrogated

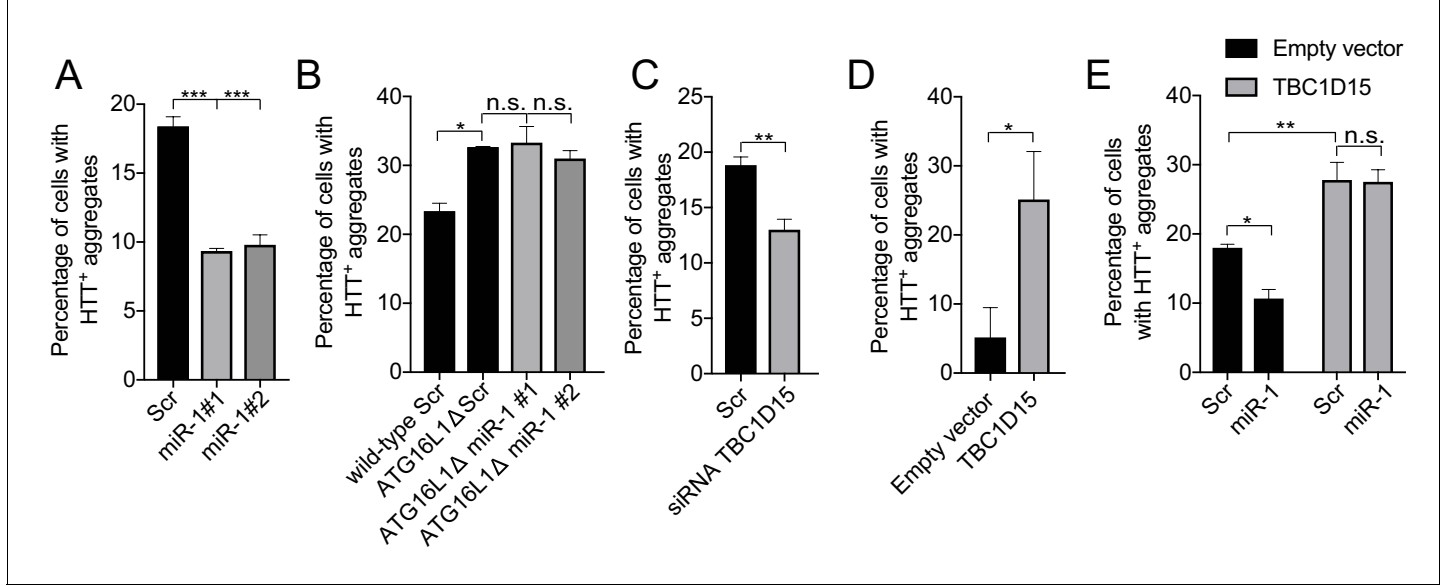

**Figure 5.** miR-1 reduces mutant Huntingtin aggregation through the autophagy pathway. (A) Quantification of the percentage of cells containing HTT-positive aggregates co-expressing scrambled (Scr) or miR-1 mimics with EGFP-$HTT_{Q74}$ for 48 hr ± SEM (n = 3, 200–400 cells per replicate). ***p<0.001 (one-way ANOVA). (B) CRISPR/Cas9 ATG16L1 knockout HeLa cells co-expressing scrambled (Scr) or miR-1 mimics with EGFP-$HTT_{Q74}$ for 48 hr. Quantification of the percentage of cells containing HTT-positive aggregates ± SEM (n = 3, 200–400 cells per replicate). *p<0.05 (Student's t-test), n.s. not significant (one-way ANOVA). (C–E) Quantification of the percentage of cells containing HTT-positive aggregates in HeLa cells co-expressing EGFP-$HTT_{Q74}$ with (C) scrambled (Scr) or siRNA against TBC1D15 for 48 hr (n = 4, 200–400 cells per replicate), (D) empty or TBC1D15 overexpression vector for 24 hr (n = 3, 200–400 cells per replicate), or (E) a combination of Scr or miR-1 mimic together with empty or TBC1D15 overexpression vector for 48 hr (n = 6, 200–400 cells per replicate) ± SEM. (C–D) *p<0.05, **p<0.005, n.s. not significant (Student's t-test) or (E) (two-way ANOVA with Dunnett's correction).

The online version of this article includes the following figure supplement(s) for figure 5:

**Figure supplement 1.** miR-1 overexpression reduces $HTT_{Q74}$ accumulation IF images of HeLa cells co-expressing scrambled miRNA (Scr) or independent miR-1 mimics with EGFP-$HTT_{Q74}$ stained with antibodies against LC3 (red), phalloidin (blue), and DAPI (gray).

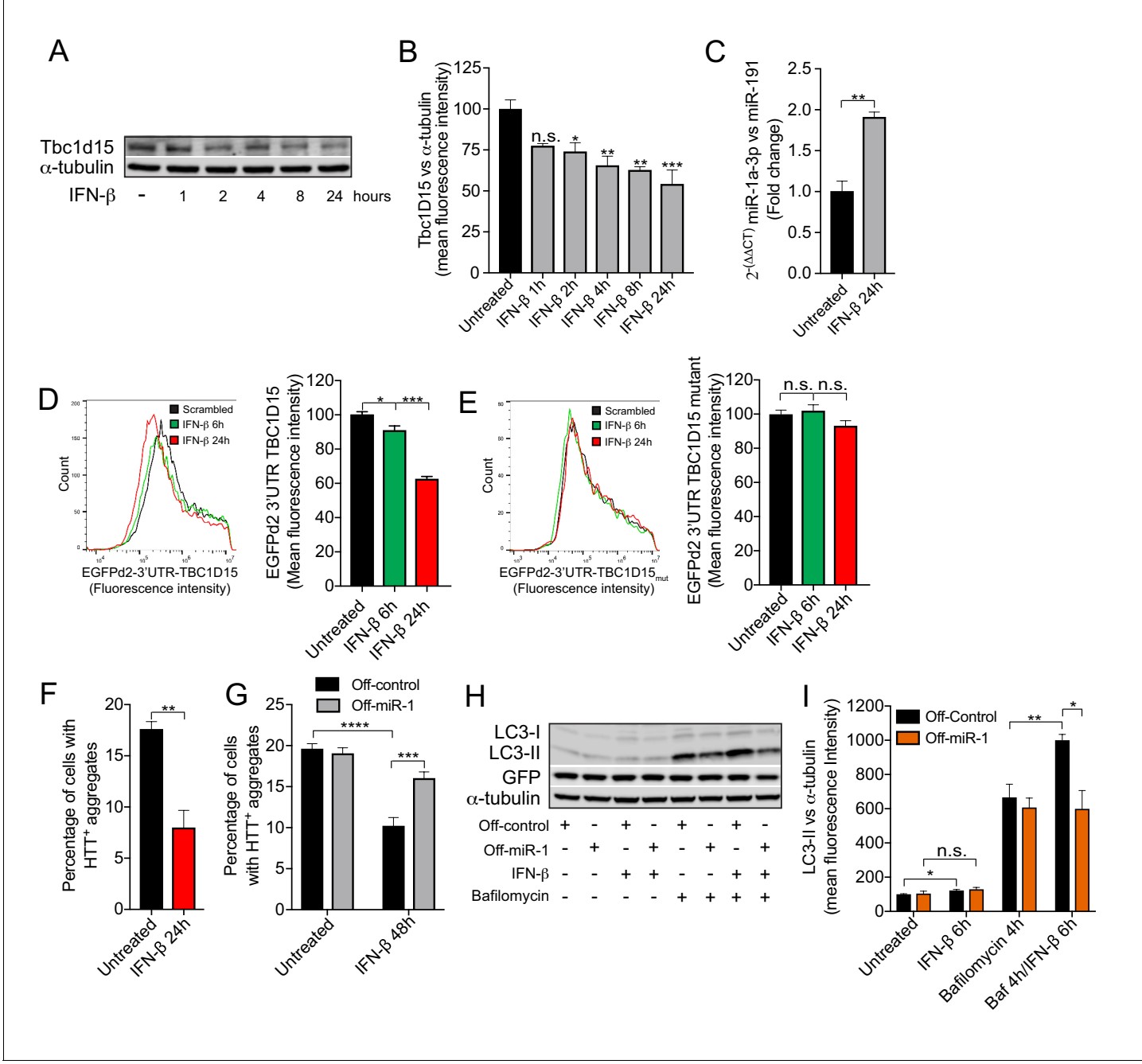

**Figure 6.** IFN-β induction of miR-1 controls mutant Huntingtin aggregation. (**A**) WB and (**B**) quantification of Tbc1d15 normalized to α-tubulin of cortical neurons from mice treated with recombinant mouse IFN-β (100 U/ml) for 1–24 hr (n = 4). Data are mean fluorescence intensities of bands ± SEM. n.s. not significant to the control, *p<0.05, **p<0.01, ***p<0.001 (one-way ANOVA). (**C**) RT-PCR of miR-1a-3p normalized to miR-191 from mouse cortical neurons treated with recombinant mouse IFN-β (100 U/ml) for 24 hr (n = 3). **p<0.01 (Student's t-test). (**D–E**) Flow cytometry analysis of HeLa cells expressing (**D**) GFPd2-3'UTR TBC1D15 (n = 4) or (**E**) mutated GFPd2-3'UTR TBC1D15$_{mutant}$ (n = 5) treated with recombinant human IFN-β (1000 U/ml) for 6 or 24 hr. Data are presented as fluorescence intensity histograms and bar graphs showing mean fluorescence intensities ± SEM. *p<0.05, ***p<0.0001 (one-way ANOVA). (**F**) Quantification of HTT$_{Q74}$ aggregates in HeLa cells expressing EGFP-HTT$_{Q74}$ treated with recombinant human IFN-β (1000 U/ml) for 24 hr. Graph shows percentage of cells containing EGFP-HTT$_{Q74}$-positive aggregates (n = 4, 400 cells per replicate) ± SEM. **p<0.01 (Student's t-test). (**G**) Quantification of HTT$_{Q74}$ aggregates in HeLa cells expressing GFP-Off-control or GFP-Off-miR-1 (miR-1 hairpin inhibitor) with EGFP-HTT$_{Q74}$ and treated with recombinant human IFN-β (1000 U/ml) for 48 hr. Graph represents percentage of cells containing EGFP-HTT$_{Q74}$-positive aggregates (n = 5, 400 cells per replicate) ± SEM. ***p<0.001, ****p<0.0001 (two-way ANOVA with Bonferroni correction). (**H**) WB of LC3, GFP and α-tubulin and (**I**) quantification of LC3-II normalized to α-tubulin from HeLa cells stably expressing GFP-Off-Control and GFP-Off-miR-1 treated with

*Figure 6 continued on next page*

*Figure 6 continued*

recombinant human IFN-β (1000 U/ml) for 6 hr, bafilomycin (400 mM) for 4 hr or in combination (n = 4) ± SEM. *p<0.05, **p<0.01, n.s. not significant (Student's t-test).

The online version of this article includes the following figure supplement(s) for figure 6:

**Figure supplement 1.** IFN-β regulates TBC1D15 expression the mouse brain.
**Figure supplement 2.** IFN-β regulates miR-1 and TBC1D15 expression in HeLa cells.
**Figure supplement 3.** TBC1D15 overexpression abrogates IFN-β-induced reduction of HTT$_{Q74}$ aggregates.

IFN-β-mediated reduction of HTT$_{Q74}$ accumulation (*Figure 6—figure supplement 3C*). We previously showed that neurons from mice lacking the *Ifnb* gene display a late-stage block in autophagy, causing accumulation of α-synuclein aggregates and Lewy bodies (*Ejlerskov et al., 2015*). Similarly, CRISPR/Cas9 knockout of the *Ifnb* gene in neuronally-differentiated N2A cells increased the number of cells with HTT$_{Q74}$ aggregates (*Figure 6—figure supplement 3D*). Our data suggest that these disease-causing phenotypes may in part be explained by dysregulation of TBC1D15.

### TBC1D15 blocks autophagy through Rab7 inactivation

The small GTPase Rab7 is an instrumental component in regulating the fusion between autophagosomes and lysosomes (*Ganley et al., 2011*). In concurrence with this role, we found that Rab7 knockdown in HeLa cells causes a significant increase in LC3-II, but when Baf A1 was added no difference was observed between Rab7 knockdown cells and scrambled siRNA control, showing that lack of Rab7 causes a late-stage block in the autophagy pathway (*Figure 7A*). To determine whether the TBC1D15-mediated autophagy block is caused by its GAP activity on Rab7, we made use of two Rab7 mutants - a constitutive active (Q67L) and a constitutive inactive (T22N) - which mimic the GTP- and GDP-bound versions of Rab7, respectively. Co-expressing TBC1D15 together with Rab7$_{wt}$ or Rab7$_{T22N}$ did not change the autophagy block induced by TBC1D15, however, when co-expressed with Rab7$_{Q67L}$, the LC3-II levels were significantly increased upon treatment with Baf A1 (*Figure 7B*). By immunofluorescence, we observed that in HeLa cells expressing high levels of TBC1D15, the distribution of Rab7$_{wt}$ was more cytosolic and there was an accumulation of large autophagosome structures, which resembles the distribution observed in cells expressing Rab7$_{T22N}$ (*Figure 7C*). Conversely, in cells with low TBC1D15 expression, Rab7$_{wt}$ was, as expected, associated with vesicular structures and without accumulation of autophagosomes. In cells expressing the constitutive active Rab7$_{Q67L}$, even high expression of TBC1D15 did not affect the vesicular distribution of Rab7 and minimal autophagosome accumulation was observed. By using an antibody that specifically recognises GTP-bound Rab7, we further confirmed by immunoprecipitation and immunofluorescence that TBC1D15 expression reduces the amount of GTP-bound Rab7 in HeLa cells (*Figure 7D–E*). Collectively, these data provide evidence that TBC1D15 acts as a GAP against Rab7 and thereby promotes Rab7 inactivation, which consequently causes a late-stage block in autophagy.

## Discussion

This study provides a previously unknown and highly conserved link between miR-1 and autophagy. Using gene knockouts and overexpression experiments, we demonstrate that elevated levels of miR-1 promote autophagy flux and reduce the accumulation of toxic protein aggregates in both *C. elegans* and mammalian cells. In *C. elegans, mir-1* functions to prevent the accumulation of polyglutamine aggregates in body wall muscle and abrogates the detrimental effects of α-synuclein and heat stress on behaviour and physiology. In mammalian cells, miR-1 promotes autophagy to protect against the accumulation of mutant huntingtin in mouse cortical neurons and HeLa cells. The mechanistic basis for the autophagy-promoting effect of miR-1 is through regulation of $\underline{T}$re-2/$\underline{B}$ub2/$\underline{C}$DC16 (TBC) Rab GTPase-activating proteins TBC-7 and TBC1D15 in *Caenorhabditis elegans* and mammalian cells, respectively. Of particular relevance is the known function of TBC1D15 in controlling the activity of Rab7, which is a central regulator of autophagy acting after LC3-II conjugation/autophagosome formation (*Gutierrez et al., 2004*; *Kirisako et al., 1999*; *Peralta et al., 2010*; *Zhang et al., 2005*). Loss of Rab7 activity causes an impairment of autophagic flux and autophagosome accumulation, the same phenotypes we observe with overexpression of TBC1D15. These data are consistent

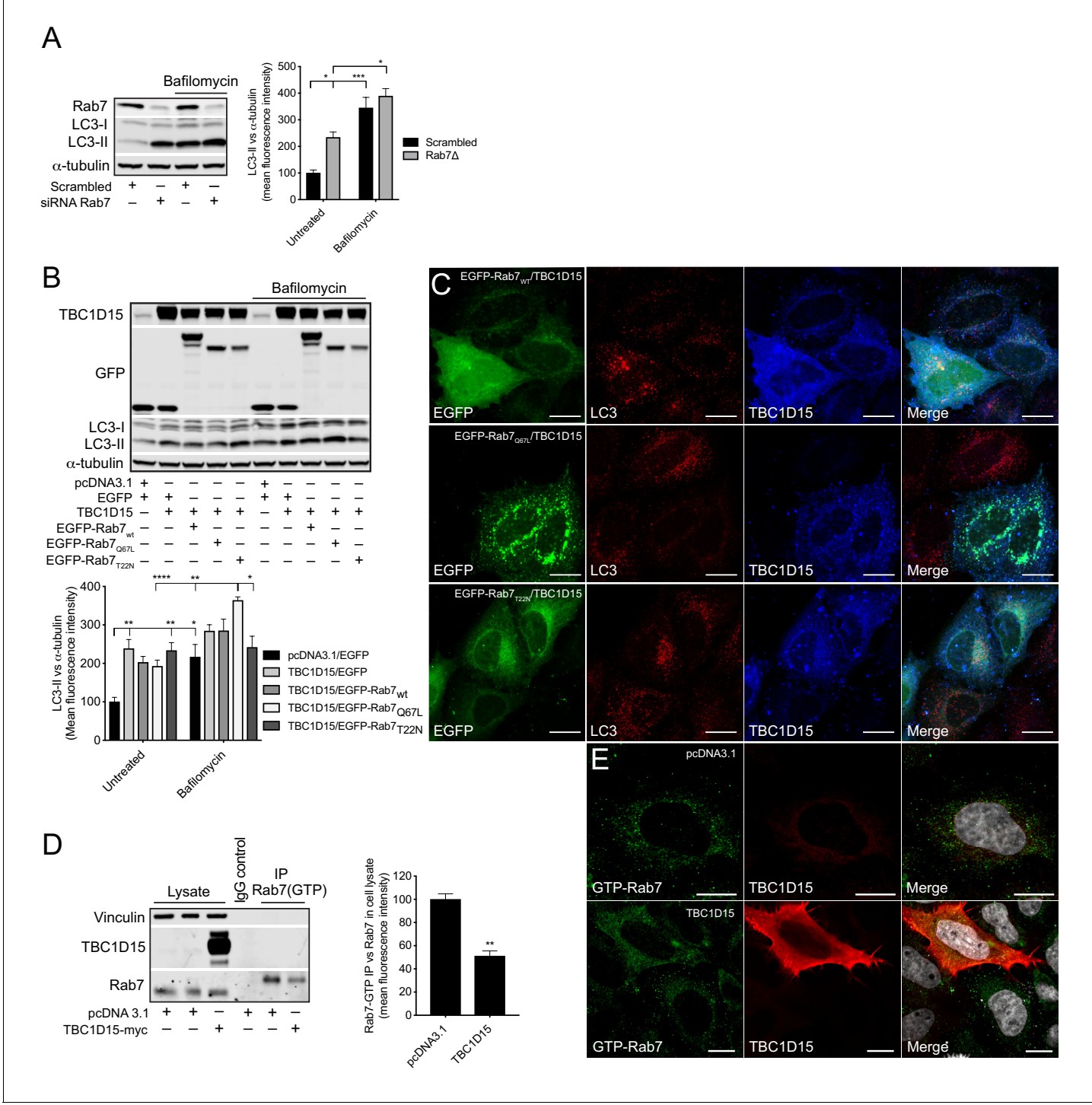

**Figure 7.** TBC1D15 reduces GTP-bound Rab7. (**A**) WB and quantification of LC3-II from HeLa cells transfected with scrambled siRNA or siRNA against Rab7 (Rab7Δ) +/- bafilomycin (4 hr, 400 nM). Data are mean fluorescence intensities of bands ± SEM normalised to α-tubulin (n = 3). *p<0.05, ***p<0.001 (two-way ANOVA). (**B**) WB and quantification of LC3-II normalised to α-tubulin from HeLa cells co-expressing pcDNA3.1 and EGFP, or TBC1D15 with either EGFP, pIRESneo-myc–Rab7wt, EGFP-Rab7Q67L, or EGFP-Rab7T22N for 24 hr before treatment with bafilomycin (4 hr, 400 nM). Data are mean fluorescence intensities of bands ± SEM normalised to α-tubulin (n = 6). *p<0.05, **p<0.01, ***p<0.001 (two-way ANOVA). (**C**) Immunofluorescence (IF) images of HeLa cells co-expressing TBC1D15 with pIRESneo-myc-Rab7wt, EGFP-Rab7Q67L or EGFP-Rab7T22N stained with antibodies against LC3 and TBC1D15. Scale bars, 10 μm. (**D**) WB showing immunoprecipitation (IP) of GTP-bound (active) Rab7 from HeLa expressing pcDNA3.1 or TBC1D15. Data are mean fluorescence intensities of GTP-bound Rab7 (IP) normalized to the endogenous level of Rab7 (cell lysate) ± SEM

*Figure 7 continued on next page*

*Figure 7 continued*

(n = 3). **p<0.01 (Student's t-test). (E)IF of HeLa cells expressing empty vector or TBC1D15 stained with antibodies against GTP-bound Rab7, TBC1D15 and DAPI. Scale bar, 10 μm.

with TBC1D15 being a GAP for Rab7, and thereby decreasing the activity of this Rab GTPase (*Gutierrez et al., 2004*; *Kirisako et al., 1999*; *Peralta et al., 2010*; *Zhang et al., 2005*). In a previous study, we showed that therapeutic application of the cytokine IFN-β can alleviate models of neurodegenerative disease (*Ejlerskov et al., 2015*). However, the molecular mechanism underlying the therapeutic effect of IFN-β was not fully defined. We found here that IFN-β induces miR-1 expression in mouse cortical neurons and that this regulation, at least in part, accounts for the therapeutic function of IFN-β in clearing aggregation-prone proteins.

This work shows that miR-1 controls the expression of the related proteins TBC-7 in worms and TBC1D15 in mammalian cells. Using genetics and mutational analysis of the *tbc-7* and TBC1D15 3'UTRs, we revealed that miR-1 directly regulates the expression of these genes. Conservation of this regulatory relationship in such evolutionarily distant species suggests that regulation of TBC protein expression by miR-1 is critical for survival. We did not however detect a detrimental effect on lifespan of *C. elegans* lacking *mir-1*, therefore, we hypothesize that miR-1 plays an essential role, at least in worms, in combating proteotoxic stress. This is supported by the sensitivity of *mir-1* mutant *C. elegans* to heat stress and the age-dependant decline of motility caused by human α-synuclein.

Our results show that miR-1 expression, and its regulation of TBC1D15 in mammalian cells, is controlled by IFN-β. How IFN-β controls miR-1 expression is unknown. In mammals, IFN-β, and IFN-α subtypes, are known to interact with the IFN α/β receptors 1 and 2, which dimerize to activate the JAK1 and TYK2 kinases (*Shuai et al., 1993*). Upon activation JAK1 and TYK2 phosphorylate and activate the family of STAT proteins, which form homo and heterodimers, translocate to the nucleus and regulate the expression of interferon-stimulated genes (*Silvennoinen et al., 1993*). It will be interesting to investigate if miR-1 is regulated by this canonical pathway to control autophagy flux. *C. elegans* does not encode IFN-β, however, STAT proteins are expressed (*Tanguy et al., 2017*; *Wang and Levy, 2006*). Intriguingly, RNAi-induced knockdown of STA-1 causes lethality to animals expressing α-synuclein in body wall muscle, suggesting that this transcription factor may control accumulation of aggregation-prone proteins in *C. elegans* (*Hamamichi et al., 2008*).

To conclude, we have identified a highly conserved regulatory axis through which the miR-1 gene controls the accumulation of aggregation-prone proteins in *C. elegans* and mammalian cells. We determined that miR-1 performs these protective roles by controlling the expression of TBC proteins - TBC-7 in worms and TBC1D15 in mammals. This conserved mechanistic relationship maintains appropriate levels of autophagic flux to enable toxic protein aggregates to be efficiently removed. Our data imply that deficits in miR-1 and TBC protein function may contribute to the etiology of protein aggregation disorders and their manipulation by IFN-β could provide novel therapeutic opportunities in treating these diseases.

# Materials and methods

## *C. elegans* and mouse strains

All *C. elegans* strains were cultured at 20°C as previously described unless otherwise stated (*Brenner, 1974*). The following strains were used: N2 (Bristol strain, wild-type), RJP3690 *mir-1(gk276)I*, RJP3691 *mir-1(n4102)I*, AM141 *rmIs133[Punc-54:: Q40::YFP]X*, RJP3636 *mir-1(gk276)I; rmIs133[Punc-54::Q40::YFP]X*, RJP3672 *mir-1(n4102)I; rmIs133[Punc-54::Q40::YFP]X*, RJP3584 *mir-80(nDf53)III; rmIs133[Punc-54::Q40::YFP]X*, RJP3596 *mir-1(gk276)I rpIs194[Pmyo-3::mir-1; Pelt-2::gfp]; rmIs133 [Punc-54::Q40::YFP]X*, RJP3636 *mir-1(gk276)I; rpEx195[Pmyo-2::mir-1; Pmyo-2::mCherry]; rmIs133 [Punc-54::Q40::YFP]X*, RJP3660 *mir-1(gk276)I; rpEx329[Pges-1::mir-1; Pmyo-2::mCherry]; rmIs133 [Punc-54::Q40::YFP]X*, RJP3657 *mir-1(gk276)I; rpEx196[Pmyo-3::mir-1*; Pmyo-2::mCherry]; rmIs133 [Punc-54::Q40::YFP]X*, NL5901 *pkIs2386[Punc-54:: α-synuclein::YFP + unc119(+)*, RJP3679 *mir-1 (gk276)I; pkIs2386[Punc-54:: α-synuclein::YFP + unc119(+)*, RJP3595 *mir-1(n4102)I; pkIs2386[Punc-54:: α-synuclein::YFP + unc119(+)*, RJP3920 *rpEx1674[Ptbc-7::GFP + Pttx-3::mCherry]*.

*Ifnb*$^{-/-}$ mice (**Erlandsson et al., 1998**) were backcrossed 20 generations to B10.RIII and housed in standard facilities. Sex- and weight-matched B10.RIII wild type mice (*Ifnb*$^{+/+}$) were used as control. Experiments that were performed in accordance with the ethical committees in Denmark and approved by the Institutional Review Boards.

## Generation of *C. elegans* transgenic strains

All constructs were injected into young adult hermaphrodites as complex arrays with *PvuII* digested bacterial DNA (80 ng/µl) and *Pmyo-2::mCherry* (5 ng/µl) or *Pelt- 2::gfp* as co-transformation marker.

## RNA-mediated interference

RNAi clones were obtained from the Ahringer *C. elegans* RNAi feeding library. All clones were sequenced and verified before use. Experiments were performed as follows; YA staged animals were moved to RNAi bacteria-seeded NGM plates and left to produce progeny for three days. Then 10 L4 staged animals were picked to plates seeded with 50 µl RNAi bacteria and left at 20°C for 24 hr. 4 plates with 10 worms were assayed for each of three replicates. Then animals were heat shocked for 5 hr at 35°C in a single layer in a ventilated incubator to ensure an equal distribution of heat. After heat shock the animals were left to recover for 17 hr at 20°C and then scored for survival by touching with a platinum wire and the animals that did not respond were scored as dead.

## Quantification of aggregates

The total number of aggregates was counted in body wall muscles using a Zeiss, AXIO Imager M2 fluorescence microscope at magnification 40x. All experiments were performed on L4 animals (for Q40::YFP) or day one adults (for α-synuclein::YFP) in triplicates with at least 10 worms counted per replicate. The experimenter was blind to genotype and the presence or absence of rescuing arrays.

## Thrashing assay

To assay motility, animals at day 3 or day seven post L4 were placed in 10 ul of M9 liquid, allowed to recover for 10 s. and then number of body bends was counted for one minute. A total of 10 worms were counted per each of three replicates. Animals not moving at all were censored from the experiments. The experimenter was blind to genotype.

## Heat shock assay

Five young adult worms were cultured on nematode growth medium (NGM) plates seeded with 300 µl of OP50 *Escherichia coli* bacteria to produce progeny at 20°C for three days. Ten L4 larval staged animals were incubated on NGM plates seeded with 50 µl of 24 hr old OP50 bacteria at 20°C for 24 hr. Four plates with ten worms were assayed for each of three replicates. Animals were heat shocked for 4 hr at 35°C in a single layer in a ventilated incubator to ensure an equal distribution of heat. After heat shock, animals were recovered for 17 hr at 20°C and scored for survival by touching with a platinum wire. Animals that did not respond were scored as dead. The experimenter was blind to genotype and the presence or absence of rescuing arrays.

## RNA preparation and qRT-PCR analysis

### *C. elegans*

RNA sequencing and RT-qPCR experiments were performed in triplicate. RNA was isolated from synchronised L4 animals: 2400 animals/sample for RNA-seq and 400 animals/sample for qPCR validation. Samples were washed three times in M9 buffer, resuspended in TRIzol (Invitrogen) and frozen in liquid nitrogen. Samples were repeatedly thawed at 37°C, vortexed for 30 s, then re-frozen in liquid nitrogen a total of 7 times. Homogenates were mixed with chloroform (Sigma), centrifuged and RNA within the upper phase was purified using the RNeasy mini kit (Qiagen) as per kit instructions, and included DNase digestion. For qPCR analysis, 300 ng of purified RNA was converted to cDNA using the ImProm II Reverse Transcription System (Promega), as per kit instructions, using an OligodT:Random primer ratio of 1:3. Samples were diluted to 5 ng/µl, qPCR analysis was performed using LightCycler 480 SYBR Green (Roche). RNA expression levels were normalized to two reference genes, *cdc-42* and *pmp-3*. The oligonucleotides used are available on request.

## Mammalian cells

For detection of miR-1–3 p and miR-191, RNA was extracted with QIAzol (Qiagen) and purified with a miRNeasy mini kit (Qiagen cat. no. 217004). TaqMan MicroRNA Reverse Transcription kit was used for cDNA synthesis (20 ng RNA from cortical neurons and 100 ng RNA from HeLa cells) using miR-specific TaqMan probes according to manufacturer description. For quantitative real-time PCR miR-1–3p and miR-191 specific TaqMan probes were used according to manufacturer description using an Applied Biosystem StepOne Plus Real-Time PCR machine for detection. Each sample was analysed in technical triplicates. For detection of TBC1D15 mRNA levels, HeLa cells were transfected with scrambled or miR-1 mimic and after two days mRNA was extracted with RNeasy mini kit (Qiagen, cat. no. 74104) and cDNA generated with QuantiTect Reverse Transcription kit using 1 µg of mRNA (Qiagen; cat. no. 205313). RT-PCR quantification was performed with the Maxima SYBR green/ROX qPCR kit (Thermo Scientific; cat. no. K0222) using 100 ng of cDNA per sample and signal was measured with an Applied Biosystem StepOne Plus RT PCR system. The following RT-PCR primers were used: human TBC1D15; Fw: GGA TGC CGA AGT AAT AGT GG; Rev: ACT GGA GTC CTT TCT AGC; human GAPDH: Fw: GAC AAC AGCCTC AAG ATC ATC; Rev: ATG AGT CCT TCC ACG ATA. Three biological replicates were measured in technical triplicates.

## RNA sequencing library construction and transcriptome analysis

RNA sequencing was performed at Micromon Genomics (Monash University). mRNA samples were converted to indexed Illumina sequencing libraries using Illumina's TruSeq Stranded mRNA Sample Prep Kit, employing oligo (dT)-conjugated beads to enrich for polyadenylated transcripts. Libraries were quantitated using a Qubit DNA HS kit (Invitrogen, Carlsbad CA., USA), sized using an AATI Fragment Analyzer (Advanced Analytical Technologies Inc, USA), and sequenced on an Illumina NextSeq500 configured to produce 75 nt paired-end reads. Fastq files were generated by bcl2fastq, trimming 3' adapter sequences.

The sequencing reads in fastq format were processed using the RNAsik pipelining tool, version 1.5.0 as follows. Reads were assessed for quality and duplication using FastQC v0.11.5 (http://www. bioinformatics.babraham.ac.uk/projects/fastqc) and mapped to the *C. elegans* genome (version WBcel235, downloaded from Ensembl) using STAR v2.5.2b (*Dobin et al., 2013*). Uniquely mapping read-pairs were assigned to annotated transcript exons (including splice-junctions) contained in the Ensembl GTF file for genome build WBcel235 using FeatureCounts v1.5.2 (*Liao et al., 2014*), aggregating at the gene level to produce genewise counts for each sample. This gene-count matrix was loaded into the Degust tool (http://degust.erc.monash.edu) for differential gene expression analysis. Genes that failed to accrue at least 10 counts in at least one sample were filtered out and the samples were normalized for library size by the TMM method (*Robinson and Oshlack, 2010*). Testing for differential gene expression between the miR-1 and N2 conditions was then performed using Limma-voom (*Law et al., 2014*).

## Molecular cloning

*mir-1* rescue constructs were generated by PCR amplification of the *mir-1* hairpin and cloned downstream of the *myo-3*, *myo-2* or *ges-1* promoters. A standard site-directed mutagenesis protocol was used to generate the *mir-1* construct with mutated seed sequence *mir-1\**. The oligonucleotides used are available on request. The human TBC1D15 3'UTR was amplified from genomic DNA and subcloned into the pCAG-GFPd2 vector, a gift from Connie Cepko lab (Addgene plasmid #14760; http:// n2t.net/addgene:14760; RRID:Addgene 79148) using *NotI* and *Bsu36I*. Forward primer: TATA TgcggccgcTCACTGTTCTTGCTTTTTGGG and reverse primer: CCATTAATTAAAATGTCTTCAGAA TGCTcctgaggGTGC. Mutations in the miR-1 seed region of the 3'UTR of TBC1D15 were introduced using a site-directed mutagenesis kit (Agilent, cat. no. 200515).

## Microscopy

Animals were anaesthetized with 20 mM NaN$_3$ on 5% agarose pads and images were taken by an AXIO Imager M2 fluorescence microscope and Zen software (Zeiss).

## Mammalian cell culture

For primary cortical neuron (CN) cultures, cortexes were dissected from 1-day-old B10.RIII mice pups and processed as previously described (*Ejlerskov et al., 2015*). Neurons were cultured in Neurobasal medium (Gibco) containing B27 (2%), GlutaMax (0.5 mM) and gentamicin (10 μg/ml) for 8–10 days on culture plates pre-coated with poly-D-lysine (137,500 cells/cm$^2$). Half of the medium was changed every 3–4 days. HeLa and N2A neuroblastoma cells were maintained in DMEM containing GlutaMax, 10% Foetal bovine serum (FBS) and 1% Penicillin/Streptomycin (P/S). Media was changed every 2–3 days and cells were split every 3–4 days. For neuronal differentiation, N2A cells were cultured for 4 days in DMEM medium containing GlutaMax, 1% FBS and retinoic acid (20 μM).

## Transfection

Cells were seeded at a density of 20,000 cells/cm$^2$ and the following day they were transfected with Lipofectamine 2000 or Mirus TransIT according to manufactures protocol. The following day cells were treated with bafilomycin A1 (400 nM) for 4 hr or were left untreated. The following plasmids were used: miRIDIAN microRNA Human hsa-miR-1–3 p – mimics (Dharmacon; cat. no. C-300585-05-0005, C-300586-05-0005); Lentivector hsa-miR-1–3 p inhibitor and control in pLenti-III-miR-Off vector from abmgood (cat. no. mh30019 and m007); mRFP-GFP-LC3 was a kind gift from Tamotsu Yoshimori; pEF6-myc-TBC1D15 a kind gift from Aimee Edinger (Addgene plasmid# 79148; http://n2t.net/addgene:79148; RRID:Addgene 79148); pEGFP-C1 (Clontech); EGFP-HTT$_{Q74}$ (vector backbone pEGFP-C1; HTT exon 1) (*Narain et al., 1999*); pCAG-GFPd2 a gift from Connie Cepko lab (Addgene plasmid #14760), pCAG-GFPd2-3′UTR-TBC1D15, pCAG-GFPd2-3′UTR-TBC1D15$_{mutant}$, pIRESneo-myc-Rab7$_{wt}$, pEGFP-C1-Rab7$_{Q67L}$, pEGFP-C1-Rab7$_{T22N}$. Rab7-expressing plasmids were kind gifts from Professor Matthew Seaman, University of Cambridge. When transfecting with miR-1 mimics or siRNA knockdown oligos the medium was changed the following day and bafilomycin A1 was added 48 hr after initial transfection. For co-transfections with miR-1 mimics cells were transfected with miR-1 24 hr prior to EGFP-HTT$_{Q74}$, TBC1D15, empty vector, GFPd2-3′UTR-TBC1D15 or GFPd2-3′UTR-TBC1D15$_{mutant}$.

## Immunofluorescence staining and imaging

After 48–72 hr of EGFP-HTT$_{Q74}$ expression, cells were fixed in 4% PFA for 10 mins, blocked in blocking buffer (5% normal goat serum, 1% bovine serum albumin, and 0.25% triton-X-100) and incubated with LC3 antibodies (Cosmo, cat.no. CAC-CTB-LC3-2-IC) dissolved 1:150 in blocking buffer overnight at 4°C. The following day, cells were washed three times in PBS, incubated with Alexa fluor secondary antibodies (Invitrogen) 1:1000 and phalloidin 633 (Molecular Probes, cat. no. A22287) 1:400 in blocking buffer for 60 mins at room temperature. Subsequently, nuclei were stained for 5 min with DAPI (Sigma, D9564) 1:1000 in PBS, washed three times in PBS, and mounted on glass slides with ProLong Gold Antifade Mountant (cat. no. P36930). Images were acquired with a Zeiss 880 confocal microscope, equipped with a live cell imaging incubator, using 405 nm, 488 nm, and 568 nm, and 633 nm lasers and a pinhole of 0.8 μm. For live cell imaging, cells were maintained at 37°C and 5% CO$_2$ in a humidified incubator and images were acquired at a speed of 1 per second for 2 min. Movies were generated as avi files in ImageJ and displayed at a speed of 10 frames per second. LC3- (in fixed cells detected with antibody) and mRFP-GFP-LC3-positive (live cell imaging) vesicles were quantified using ImageJ and Volocity. Each image contained 2–4 cells and a total of 12–24 cells were scored in each technical replicate. EGFP-HTT$_{Q74}$ positive aggregates were quantified by manual counting using the 63x objective on a Zeiss Axio Imager M2 microscope. Each condition was set up in duplicates and 200–400 cells were counted per technical replicate.

## Flow cytometry analysis

HeLa cells were transfected with scrambled or miR-1 mimic one day prior to transfection with GFPd2-3′UTR-TBC1D15 or GFPd2-3′UTR-TBC1D15$_{mutant}$. The following day, cells were placed on ice and stained with live/dead cell marker (Invitrogen, cat. no. L23102) according to manufactures protocol before analysis with Accuri C6 flow cytometer using the FL1-A channel for detection of GFP signal and FL4-A for detection of dead cells. The cells were analysed in technical duplicates measuring the mean intensity fluorescence for 15,000–20,000 cells per well in the live cell population.

## Western blot analysis

### C. elegans

Young adult animals were picked directly into SDS sample buffer, boiled for 15 mins at 95°C and cooled on ice. The solution was centrifuged for 10 mins at 3000 rpm and equal amounts of sample were loaded onto a protein gel, separated by SDS page and protein analysis was assayed with GFP antibody (Roche), PolyUbiquitin antibody (Sigma) and α-tubulin (12G10 - Developmental Studies Hybridoma Bank, University of Iowa).

### Mammalian cells

Cells were lysed in Triton X-100 buffer (1% Triton X-100, 100 mM NaCl, 50 mM Tris-HCl, 1 mM EGTA, and 10 mM $MgCl_2$) containing phosphatase inhibitor cocktail 2 and 3 (Sigma, cat.no. P5726 and P0044), and cOmplete protease inhibitor cocktail (Roche, cat. no. 11873580001) for five mins at room temperature and then kept on ice. The lysis suspensions were harvested and centrifuged at 16,100 g for 5 mins at 4°C and the supernatants collected. For primary tissue isolation, brains from 3 months old wt and $Ifnb^{-/-}$ mice were dissected and homogenized in Triton-X-100 lysis buffer (same buffer as detailed above) with a 1 ml Dounce homogenizer. Lysates were spun through a Qiagen Shredder tube at 16,100 g for 5 min at 4°C and the supernatants were harvested and spun an additional round at 16,100 g for 10 min at 4°C, before the final supernatants were collected. The protein concentrations were measured with Biorad DC protein assay (cat. no. 5000112) reagent and equal amounts of sample were run on SDS-PAGE gels and transferred to PVDF Immobilon-FV membranes (Millipore, cat. no. IPFL20200). Membranes were blocked in either 5% milk or 5% BSA and subsequently incubated with primary antibodies overnight. The following day membranes were washed three times in PBS with 0.1% tween-20 (PBS-T), and incubated with species-specific secondary antibodies coupled to 680 nm or 800 nm fluorophores (Li-Cor) in 5% milk or 2% BSA. Finally, the membranes were washed three times in PBS-T and signal detected in Li-Cor Odyssey scanner using the 700 nm and 800 nm emission filters. Mean fluorescence band intensities were quantified using Image Studio Lite version 5.2.

## Immunoprecipitation (IP) of GTP-bound Rab7

HeLa cells were plated in 10 cm dishes and transfected with empty vector or TBC1D15-expressing vector using Lipofectamine 2000. The following day cells were processed with Rab7 activation assay kit according the manufacturers protocol (NewEast Biosciences cat. no. 82501). In short, the cells were washed two times in ice-cold PBS⁻, resuspended in pre-warmed growth media (37°C) containing GTP-γS (100 μM), and incubated at 37°C for 30 min with gentle agitation. The cells were then lysed at 4°C for 10 min and then centrifuged at 16,100 g for 5 min at 4°C. Supernatants were collected and incubated with rotation with an antibody against GTP-bound Rab7 (cat. no. 26923) or mouse IgG negative control (DAKO cat. no. X-0931) together with protein A/G agarose beads for one hour at 4°C. Finally, the beads were washed three times in ice-cold lysis buffer and bound GTP-Rab7 was removed from the beads by boiling the samples for 5 min in 2X SDS-PAGE sample buffer. The beads were spun at 5,000 g for 10 s and the IP pull-down supernatants were harvested and processed for western blotting as described above with an antibody against Rab7 (cat. no. 21069).

## Autophagy experiments

### C. elegans

Animals were incubated at 20°C for two generations prior to the experiment. Young adult animals were then heat shocked for 1 hr at 35°C (heat shock) or recovered for 1 hr at 15°C (heat shock + recovery) before being imaged by confocal microscopy.

### Mammalian cells

For autophagy flux assays, cells were treated with bafilomycin A1 (400 nM) for 4 hr and subsequently lysed in Triton X-100 buffer and processed for WB as described above. When co-treating HeLa cells with recombinant human IFN-β, cells were pre-treated with IFN-β (1000 U/ml) for 2 hr before the addition of bafilomycin (400 nM) for 4 hr.

## Statistical analysis

All statistical analysis was performed using the GraphPad Prism 5.0 software. Student's *t*-test, one-way or two-way ANOVA analysis followed by Dunnett's or Bonferroni's multiple comparison tests were used. Data is presented as means ± SEM.

## Acknowledgements

We thank members of the Pocock and Rubinsztein laboratories for comments on the manuscript, Monash Micromon (RNA sequencing) and Stuart Archer from the Monash Bioinformatics Platform (Bioinformatics). Some strains were provided by the *Caenorhabditis* Genetics Center (University of Minnesota), which is funded by NIH Office of Research Infrastructure Programs (P40 OD010440). This work was supported by The Danish Council for Independent Research (DFF-6110–00658 to SI-N) and Lundbeck Foundation (R223-2016-849 to SI-N), UK Dementia Research Institute at the University of Cambridge (funded by the MRC, Alzheimer's Research UK and the Alzheimer's Society), the National Institute for Health Research Cambridge Biomedical Research Centre, the Wellcome Trust (095317/Z/11/Z), the Spoelberch Foundation and an anonymous donation to the Cambridge Centre for Parkinson-Plus to DCR, NHMRC (Senor Research Fellowship GNT1137645 to RP), veski Innovation Fellowship (VIF23 to RP), The Danish Council for Independent Research (DFF - 6110–00461 to PE), Lundbeck Foundation (R210-2015-3372 to PE), and Parkinsonforening in Denmark (to PE). The views expressed are those of the author(s) and not necessarily those of the NHS, the NIHR or the Department of Health and Social Care.

## Additional information

### Funding

| Funder | Grant reference number | Author |
|---|---|---|
| National Health and Medical Research Council | GNT1137645 | Roger Pocock |
| Lundbeckfonden | R223-2016-849 | Shohreh Issazadeh-Navikas |
| Lundbeckfonden | R210-2015-3372 | Patrick Ejlerskov |
| Wellcome | 095317/Z/11/Z | David C Rubinsztein |
| Det Frie Forskningsråd | DFF-6110-00461 | Patrick Ejlerskov |
| Veski | VIF23 | Roger Pocock |
| Det Frie Forskningsråd | DFF-6110–00658 | Shohreh Issazadeh-Navikas |

The funders had no role in study design, data collection and interpretation, or the decision to submit the work for publication.

### Author contributions

Camilla Nehammer, Roger Pocock, Conceptualization, Data curation, Formal analysis, Supervision, Funding acquisition, Validation, Investigation, Methodology, Project administration; Patrick Ejlerskov, Sandeep Gopal, Conceptualization, Data curation, Formal analysis, Validation, Investigation, Methodology; Ava Handley, Leelee Ng, Conceptualization, Data curation, Formal analysis, Investigation, Methodology; Pedro Moreira, Huikyong Lee, Data curation, Formal analysis, Investigation; Shohreh Issazadeh-Navikas, Data curation, Formal analysis, Supervision, Investigation; David C Rubinsztein, Conceptualization, Supervision, Funding acquisition, Project administration

### Author ORCIDs

Ava Handley https://orcid.org/0000-0003-1543-1551
David C Rubinsztein https://orcid.org/0000-0001-5002-5263
Roger Pocock https://orcid.org/0000-0002-5515-3608

Decision letter and Author response
Decision letter https://doi.org/10.7554/eLife.49930.sa1
Author response https://doi.org/10.7554/eLife.49930.sa2

## Additional files

### Supplementary files

• Supplementary file 1. Predicted *C. elegans* *mir-1* target genes. Predicted *mir-1* target genes (TargetScanWorm release 6.2) showing the number and type of putative conserved *mir-1* binding sites in their 3'UTRs. Aggregate $P_{CT}$ = probability of conserved targeting.

• Supplementary file 2. RNA sequencing data. Differentially expressed genes in *mir-1(gk276)* animals compared to wild-type for each of the three biological replicate samples. Raw counts and counts per million reads (CPM) are shown. The false discovery cut-off was set at 0.1 and absolute log fold change set to 0.3 (1.2x change in expression). Full dataset is located at NCBI - GSE128968.

• Transparent reporting form

### Data availability

RNA sequencing data have been deposited in GEO under accession code GSE128968.

The following dataset was generated:

| Author(s) | Year | Dataset title | Dataset URL | Database and Identifier |
|---|---|---|---|---|
| Archer SK, Nehammer C, Ejlerskov P, Gopal S, Handley A, Ng L, Moreira P, Rubinsztein DC, Pocock R | 2019 | Transcriptome analysis of mir-1-deficient and ins-1-deficient *Caenorhabditis elegans* | https://www.ncbi.nlm.nih.gov/geo/query/acc.cgi?acc=GSE128968 | NCBI Gene Expression Omnibus, GSE128968 |

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
