## [Decision Letter]

**Acceptance summary:**

We found the results presented in this work now convincingly support your model that the microRNA miR-1 prevents the accumulation of toxic protein aggregates by downregulating expression of TBC-7/TBC1D15 (RAB7 GEF) and thereby stimulating autophagic flux. Together with the finding that the cytokine IFN-β induces miR-1 expression, this study implicates the IFN-β-miR-1-TBC1D15 axis as a potential target for the therapy of neurodegenerative disorders.

**Decision letter after peer review:**

Thank you for submitting your article "Interferon-β-induced miR-1 alleviates toxic protein accumulation by controlling autophagy" for consideration by *eLife*. Your article has been reviewed by two peer reviewers, and the evaluation has been overseen by Hitoshi Nakatogawa as the Reviewing Editor and Huda Zoghbi as the Senior Editor.

The reviewers have discussed the reviews with one another and the Reviewing Editor has drafted this decision to help you prepare a revised submission.

Summary:

In this study, the authors propose a novel mechanism that prevents the accumulation of toxic protein aggregates by controlling autophagy. First, using worm genetics and the mammalian cell culture system, they discovered an evolutionally conserved pathway in which the microRNA miR-1 suppresses expression of the Rab GTPase-activating proteins TBC-7/TBC1D15, leading to the prevention of autophagosome-lysosome fusion. Consistently, toxic protein aggregates, which are eliminated by autophagy, accumulate in cells defective in this pathway. In addition, they found that the cytokine IFN-β induces miR-1 expression and thereby suppresses TBC1D15 expression and the accumulation of protein aggregates in mammalian cells. Thus, modulation of IFN-β/miR-1/TBC1D15 pathway may be beneficial for the therapy of neurodegenerative disorders.

Overall, the manuscript is well written and the data basically support the authors' conclusions. However, some additional experiments will be needed to strengthen their conclusions.

Essential revisions:

As essential revisions, we request the authors to address the following issues.

1) In Figures 4 and 6 as well as supplementary figures, the authors examined autophagic activity in mammalian cells based on levels of LC3-II. However, to analyze autophagy flux in the cell culture system, both the protein and mRNA levels of p62/SQSTM1 should also be examined (checking mRNA levels will be required, because p62 transcription is known to be induced under stress conditions).

2) In Figure 3A, the quality of the images is too poor. It is hard to see whether autophagy is induced based on the present images. Increasing image resolution and counterstaining with lysosomal markers will be needed.

3) In Figure 1, the authors described that "loss of mir-1 causes ~50% reduction in motility, presumably through toxicity caused by α-synuclein inclusions". As it is well known that α-synuclein hardly forms aggregates in cultured cells without aggregation seeds, the authors should show whether α-synuclein aggregates are indeed formed in the worm.

4) The authors should show how miR-1 attenuates TBC1D15 expression. As shown in Figure 2A, miR-1 KO results in the accumulation of tbc-7 mRNA. Does miR-1 binding to 3'UTR of TBC1D15 destabilize TBC1D15 mRNA? The authors should determine whether the level of TBC1D15 mRNA is reduced or unchanged, when miR1 is overexpressed in cultured cells.

5) The authors should show that TBC1D15 indeed blocks autophagosome-lysosome fusion by inhibiting Rab7 in their experimental systems.

---

## [Author Response]

Essential revisions:As essential revisions, we request the authors to address the following issues.1) In Figures 4 and 6 as well as supplementary figures, the authors examined autophagic activity in mammalian cells based on levels of LC3-II. However, to analyze autophagy flux in the cell culture system, both the protein and mRNA levels of p62/SQSTM1 should also be examined (checking mRNA levels will be required, because p62 transcription is known to be induced under stress conditions).

We have now tested the p62/SQSTM1 level by western blot and RT-PCR in HeLa cells expressing scrambled miRNA or a miR-1 mimic and we observe an increase in p62/SQSTM1 at the protein level but also a corresponding increase at the mRNA level (see Author response image 1). Thus, p62/SQSTM1 is not a suitable marker for autophagy flux in our system because of the transcriptional increase in p62/SQSTM1 we observe. As the reviewer also notes, p62/SQSTM1 is transcriptionally regulated by stress but is also involved in autophagy-independent actions, and therefore does not always follow autophagy flux (Sánchez-Martín et al. J Cell Sci. 2018 Nov 5;131(21). doi: 10.1242/jcs.222836). This is why we analyzed autophagy flux with the mRFP-GFP-LC3 construct as well as treating HeLa cells with bafilomycin, which in combination enables us to conclude that miR-1 is indeed promoting autophagy flux. In addition, we used the well-known aggregation-prone autophagy substrate HTT_Q74_, and here we show that miR-1 expression reduces the number of cells containing protein aggregates. Further, when using autophagy null cells (ATG16 CRISPR/Cas9 knock-out cells) this effect is abrogated – these data show that the effects we observe are autophagy-dependent (and numerous previous studies have used this assay or variants to assess autophagic flux – see Figure 5). These results support that miR-1 is promoting autophagy flux and thereby decrease the amount cytotoxic protein aggregates.

**Author response image 1. respfig1:** HeLa expressing scrambled or miR-1. (**A**) WB of SQSTM1/p62 and (**B**) qPCR of SQSTM1/p62.

2) In Figure 3A, the quality of the images is too poor. It is hard to see whether autophagy is induced based on the present images. Increasing image resolution and counterstaining with lysosomal markers will be needed.

The GFP::LGG-1 reporter we used for this analysis is widely used to study autophagy in *C. elegans*. The images we show in Figure 3A are comparable to those published in other studies (Chang et al., 2017). Nonetheless, we have now used an independent autophagy reporter to confirm our results. This dual fluorescence reporter (mCherry::GFP::LGG-1) has previously been validated to monitor autolysosome number by counting red vesicles as GFP is quenched in the acidic autolysosomal environment (Chang et al., 2017). Using this reporter, we show that mir-1 mutant animals are unable to mount a autophagic response to heat stress as we previously showed in Figure 3A. These new data are shown in Figure 3—figure supplement 1.

3) In Figure 1, the authors described that "loss of mir-1 causes ~50% reduction in motility, presumably through toxicity caused by α-synuclein inclusions". As it is well known that α-synuclein hardly forms aggregates in cultured cells without aggregation seeds, the authors should show whether α-synuclein aggregates are indeed formed in the worm.

The α-synuclein;:YFP reporter we used has been previously shown to accumulate inclusions in an age-dependent manner in *C. elegans* (van Ham et al., 2008). Therefore, we have now monitored α-synuclein;:YFP and found that mir-1 mutant animals have a modest yet significant increase in inclusion number compared to wild type. These new data are included in Figure 1—figure supplement 3. We however do not discount that the loss of motility may be the result of accumulation of other intermediate species.

4) The authors should show how miR-1 attenuates TBC1D15 expression. As shown in Figure 2A, miR-1 KO results in the accumulation of tbc-7 mRNA. Does miR-1 binding to 3'UTR of TBC1D15 destabilize TBC1D15 mRNA? The authors should determine whether the level of TBC1D15 mRNA is reduced or unchanged, when miR1 is overexpressed in cultured cells.

We have addressed this in Figure 2—figure supplement 5A, showing that miR-1 expression causes a mild but significant reduction in TBC1D15 mRNA.

5) The authors should show that TBC1D15 indeed blocks autophagosome-lysosome fusion by inhibiting Rab7 in their experimental systems.

We have extensively addressed this relevant issue by the following experiments which has become a new Figure 7 in the manuscript.

a) We validated that Rab7 is essential for autophagy flux by siRNA knock-down of Rab7 (Figure 7A). This causes an increase in LC3-II but in the presence of bafilomycin there is no difference between scrambled control and Rab7Δ cells, which together demonstrates a late-stage block in autophagy.

b) In order to circumvent the TBC1D15-mediated autophagy block, we made use of two Rab7 mutants, a constitutive active Rab7_Q67L_ and a constitutive inactive Rab7_T22N_, which mimic GTP- and GDP-bound Rab7, respectively. By co-expression of TBC1D15 with Rab7_wt_ and Rab7_T22N_ the TBC1D15-mediated autophagy block is not affected, however, when co-expressing with the constitutive active Rab7_Q67L_, which is not affected by the GAP activity of TBC1D15, LC3-II reaches a significant higher level in the presence of bafilomycin A1 when compared to the control sample. This shows that active Rab7 overcomes the TBC1D15-mediated autophagy block and is downstream of TBC1D15 – see Figure 7B.

c) In addition, we show by immunoprecipitation and immunofluorescence, using an antibody specific for GTP-bound Rab7, that TBC1D15 reduces the amount of GTP-bound Rab7 – see Figure 7D.

Collectively, these new data demonstrate that TBC1D15 targets and reduces the amount of active Rab7 and thereby blocks autophagy flux.